# EFFICIENT MODULATION FOR VISION NETWORKS

**Xu Ma[1], Xiyang Dai[2], Jianwei Yang[2], Bin Xiao[2], Yinpeng Chen[2], Yun Fu[1], Lu Yuan[2]**
[1]Northeastern University    [2]Microsoft

## ABSTRACT

In this work, we present efficient modulation, a novel design for efficient vision networks. We revisit the modulation mechanism, which operates input through convolutional context modeling and feature projection layers, and fuses features via element-wise multiplication and an MLP block. We demonstrate that the modulation mechanism is particularly well suited for efficient networks and further tailor the modulation design by proposing the efficient modulation (EfficientMod) block, which is considered the essential building block for our networks. Benefiting from the prominent representational ability of modulation mechanism and the proposed efficient design, our network can accomplish better trade-offs between accuracy and efficiency and set new state-of-the-art performance in the zoo of efficient networks. When integrating EfficientMod with the vanilla self-attention block, we obtain the hybrid architecture which further improves the performance without loss of efficiency. We carry out comprehensive experiments to verify EfficientMod's performance. With fewer parameters, our EfficientMod-s performs **0.6 top-1 accuracy better than EfficientFormerV2-s2 and is 25% faster on GPU, and 2.9 better than MobileViTv2-1.0 at the same GPU latency.** Additionally, our method presents a notable improvement in downstream tasks, outperforming EfficientFormerV2-s by **3.6 mIoU on the ADE20K benchmark**. Code and checkpoints are available at https://github.com/ma-xu/EfficientMod.

## 1 INTRODUCTION

Vision Transformers (ViTs) (Dosovitskiy et al., 2021; Liu et al., 2021; Vaswani et al., 2017) have shown impressive accomplishments on a wide range of vision tasks and contributed innovative ideas for vision network design. Credited to the self-attention mechanism, ViTs are distinguished from conventional convolutional networks by their dynamic properties and capability for long-range context modeling. However, due to the quadratic complexity over the number of visual tokens, self-attention is neither parameter- nor computation-efficient. This inhibits ViTs from being deployed on edge or mobile devices and other real-time application scenarios. To this end, some attempts have been made to employ self-attention within local regions (Liu et al., 2021; Chen et al., 2022a) or to selectively compute informative tokens (Rao et al., 2021; Yin et al., 2022) to reduce computations. Meanwhile, some efforts (Mehta & Rastegari, 2022; Chen et al., 2022b; Graham et al., 2021) attempt to combine convolution and self-attention to achieve desirable effectiveness-efficiency trade-offs.

Most recently, some works (Liu et al., 2022b; Yu et al., 2022a; Trockman & Kolter, 2022) suggest that a pure convolutional network can also attain satisfying results compared with self-attention. Among these, FocalNet (Yang et al., 2022) and VAN (Guo et al., 2023), which are computationally efficient and implementation-friendly, show cutting-edge performance and significantly outperform ViT counterparts. Generally, both approaches consider context modeling using a large-kernel convolutional block and modulate the projected input feature using element-wise multiplication (followed by an MLP block), as shown in Fig. 1b. Without the loss of generality, we refer to this design as *Modulation Mechanism*, which exhibits promising performance and benefits from the effectiveness of convolution and the dynamics of self-attention. Although the modulation mechanism provides satisfactory performance and is theoretically efficient (in terms of parameters and FLOPs), it suffers unsatisfying inference speed when the computational resource is limited. The reasons are two-fold: $i$) redundant and isofunctional operations, such as successive depth-wise convolutions and redundant linear projections take up a large portion of operating time; $ii$) fragmentary operations in the context modeling branch considerably raise the latency and are in contravention of *guidance G3* in ShuffleNetv2 (Ma et al., 2018).

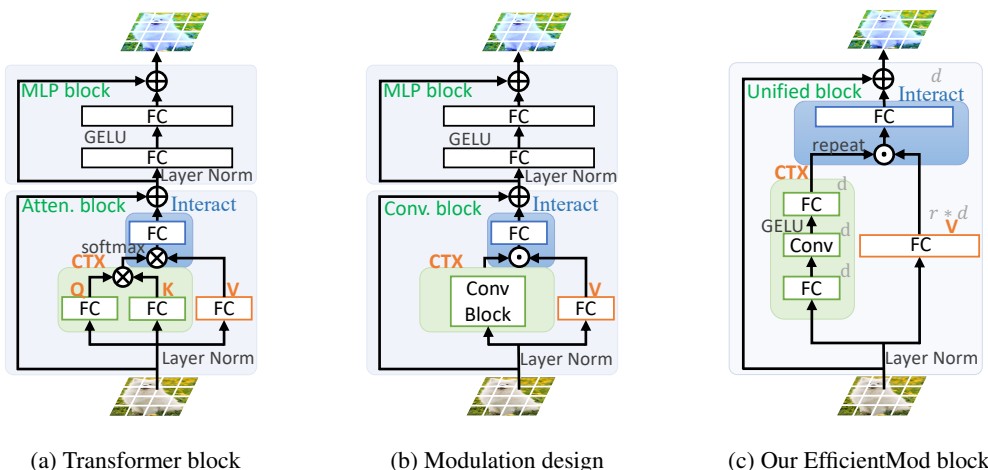

(a) Transformer block    (b) Modulation design    (c) Our EfficientMod block

Figure 1: Comparison of Transformer, abstracted modulation design, and our EfficientMod block. $\odot$ is element-wise multiplication and $\otimes$ means matrix multiplication. Compared to Transformer and abstracted modulation, our unified block efficiently modulates the projected values ($V$) via a simple context modeling design (CTX). Dimension number is indicated in (c) to aid comprehension.

In this work, we propose *Efficient Modulation*, a simple yet effective design that can serve as the essential building block for efficient models (see Fig. 1c). In comparison to modulation blocks behind FocalNet (Yang et al., 2022) and VAN (Guo et al., 2023), the efficient modulation block is more simple and inherits all benefits (see Fig. 1b and Fig. 1c). In contrast to the Transformer block, our EfficientMod's computational complexity is linearly associated with image size, and we emphasize large but local interactions, while Transformer is cubically correlated with the token number and directly computes the global interactions. As opposed to the inverted residual (MBConv) block (Sandler et al., 2018), which is still the *de facto* fundamental building block for many effective networks, our solution uses fewer channels for depth-wise convolution and incorporates dynamics (see Table 6 for comparison). By analyzing the connections and differences between our and these designs, we offer a deep insight into where our effectiveness and efficiency come from. By examining the similarities and distinctions, we gain valuable insights into the efficiency of our approach.

With our Efficient Modulation block, we introduce a new architecture for efficient networks called EfficientMod Network. EfficientMod is a pure convolutional-based network and exhibits promising performance. Meanwhile, our proposed block is orthogonal to the traditional self-attention block and has excellent compatibility with other designs. By integrating attention blocks with our EfficientMod, we get a hybrid architecture, which can yield even better results. Without the use of neural network searching (NAS), our EfficientMod offers encouraging performance across a range of tasks. Compared with the previous state-of-the-art method EfficientFormerV2 (Li et al., 2023b), EfficientMod-s outperforms EfficientFormerV2-S2 by **0.3 top-1 accuracy and is 25% faster on GPU**. Furthermore, our method substantially surpasses EfficientFormerV2 on downstream tasks, outperforming it by **3.6 mIoU on the ADE20K semantic segmentation benchmark** with comparable model complexity. Results from extensive experiments indicated that the proposed EfficientMod is effective and efficient.

## 2    RELATED WORK

**Efficient ConvNets Designs.**    One of the most profound efficient networks is MobileNet (Howard et al., 2017), which decouples a conventional convolution into a point-wise and a depth-wise convolution. By doing so, the parameter number and FLOPs are radically reduced, and the inference speed is substantially boosted. Subsequently, MobileNetV2 (Sandler et al., 2018) further pushed the field by introducing the inverted bottleneck block (also known as the MBConv block), which is now the *de facto* fundamental building block for most efficient networks (Tan & Le, 2019; Guo et al., 2022; Li et al., 2021; Peng et al., 2021; Tu et al., 2022). In addition, some other contributions are also noteworthy. Network architecture search (NAS) can provide better network designs like MobileNetv3 (Howard et al., 2019), EfficientNet (Tan & Le, 2019; 2021), and FBNet (Wu et al., 2019; Wan et al., 2020; Dai et al., 2021), *etc*. ShuffleNet (Zhang et al., 2018) leverages group operation and

channel shuffling to save computations. ShuffleNetv2 (Ma et al., 2018), FasterNet (Chen et al., 2023), and GhostNet (Han et al., 2020a) emphasize the effectiveness of feature re-use. Regarding efficient design, our model is similar to the MBConv block, but the inner operations and mechanisms behind it are different. Regarding design philosophy, our model is similar to FocalNet Yang et al. (2022) and VAN Guo et al. (2023), but is considerably more efficient and elegant. We discuss the connections and differences in Sec. 3.5 and Table 6.

**Transformers in Efficient Networks.** Transformer has garnered considerable interest from the vision community, which undoubtedly includes effective networks. Some methods, such as Mobile-Former (Chen et al., 2022b), contemplate adding self-attention to ConvNets to capture local and global interactions concurrently. Self-attention, however, endures high computational costs due to the quadratic complexity of the number of visual tokens. To eschew prohibitive computations, EfficientFormerV2 and EdgeNeXt (Maaz et al., 2023) consider MBConv blocks in the early stages and employ self-attention in the later stages when the token number (or feature resolution) is small. In contrast to earlier efforts, which combined self-attention and MBConv block to achieve a trade-off between efficiency and effectiveness, we distilled the inherent properties of self-attention, dynamics, and large receptive field and introduced these properties to our EfficientMod. We also explore the hybrid architecture that integrates self-attention and EfficientMod for better performance.

**Discussion on Efficient Networks.** Although parameter number and FLOPs are widely employed metrics to assess the theoretical complexity of a model, they do not reflect the network's real-time cost, as endorsed in ShuffleNetv2 (Ma et al., 2018). Practical guidelines for efficient network design are critical, like fewer network fragments (Ma et al., 2018) and consistent feature dimension (Li et al., 2022), *etc*. FasterNet (Chen et al., 2023) also demonstrates that low FLOPs do not necessarily lead to low latency due to inefficient low floating-point operations per second. In this work, we present EfficientMod and incorporate prior observations into our design to achieve practical effectiveness.

## 3 METHOD

### 3.1 REVISIT MODULATION DESIGN

We first derive the general concept of modulation mechanism from VAN and FocalNet.

**Visual Attention Networks.** VAN (Guo et al., 2023) considers a convolutional attention design, which is simple yet effective. Specifically, given input feature $x \in \mathbb{R}^{c \times h \times w}$, we first project $x$ to a new feature space using a fully-connected (FC) layer (with activation function) $f(\cdot)$ and then feed it into two branches. The first branch $\mathtt{ctx}(\cdot)$ extracts the context information, and the second branch is an identical mapping. We use element-wise multiplication to fuse the feature from both branches, and a new linear projection $p(\cdot)$ is added subsequently. In detail, a VAN block can be written as:

$$\text{Output} = p\left(\mathtt{ctx}\left(f\left(x\right)\right) \odot f\left(x\right)\right), \tag{1}$$

$$\mathtt{ctx}\left(x\right) = g\left(\text{DWConv}_{7,3}\left(\text{DWConv}_{5,1}\left(x\right)\right)\right), \tag{2}$$

where $\odot$ is element-wise multiplication, $\text{DWConv}_{k,d}$ means a depth-wise convolution with kernel size $k$ and dilation $d$, and $g(\cdot)$ is another FC layer in the context branch. Following the design philosophy of MetaFormer (Yu et al., 2022a), the VAN block is employed as a token-mixer, and a two-layer MLP block (with a depth-wise convolution) is adjacently connected as a channel-mixer.

**FocalNets.** FocalNets (Yang et al., 2022) introduced the Focal Modulation that replaces self-attention but enjoys the dynamics and large receptive fields. FocalNet also considers a parallel two branches design, where one context modeling branch $\mathtt{ctx}(\cdot)$ adaptively aggregates different levels of contexts and one linear project branch $v(\cdot)$ project $x$ to a new space. Similarly, the two branches are fused by element-wise multiplication, and an FC layer $p(\cdot)$ is employed. Formally, the hierarchical modulation design in FocalNet can be given by (ignoring the global average pooling level for clarity):

$$\mathtt{ctx}\left(x\right) = g\left(\sum_{l=1}^{L} \text{act}\left(\text{DWConv}_{k_l}\left(f\left(x\right)\right) \odot \mathtt{z}\left(f\left(x\right)\right)\right)\right), \tag{3}$$

where $\mathtt{ctx}$ includes $L$ levels of context information that are hierarchically extracted by depth-wise convolutional layer with a kernel size of $k_l$, $z(\cdot)$ project $c$-channel feature to a gating value. $\text{act}(\cdot)$ is GELU activation function after each convolutional layer.

**Abstracted Modulation Mechanism.** Both VAN and FocalNet demonstrated promising representational ability and exhibited satisfying performance. By revisiting as aforementioned, we reveal that both methods share some indispensable designs, which greatly contribute to their advancements. Firstly, the two parallel branches are operated individually, extracting features from different feature spaces like self-attention mechanism (as shown in Fig. 1a). Secondly, for the context modeling, both considered large receptive fields. VAN stacked two large kernel convolutions with dilation while FocalNet introduced hierarchical context aggregation as well as a global average pooling to achieve a global interaction. Thirdly, both methods fuse the features from two branches via element-wise multiplication, which is computationally efficient. Lastly, a linear projection is employed after feature fusion. We argue that the gratifying performance of the two models can be credited to the above key components. Meanwhile, there are also distinct designs, like the particular implementations of context modeling and the design of feature projection branches (shared or individual projection). Consolidating the aforementioned similarities and overlooking specific differences, we abstract the modulation mechanism as depicted in Fig. 1b and formally define the formulation as:

$$\text{Output} = p\left(\text{ctx}\left(x\right) \odot v\left(x\right)\right). \tag{4}$$

The abstracted modulation mechanism inherits desirable properties from both convolution and self-attention but operates in a convolutional fashion with satisfying efficiency in theory. Specifically, Eq. 4 enjoys dynamics like self-attention due to the element-wise multiplication. The context branch also introduces local feature modeling, but a large receptive field is also achieved via large kernel size (which is not a bottleneck for efficiency). Following VAN and FocalNet, a two-layer MLP block is constantly introduced after the modulation design, as shown in Fig. 1c. Besides aforementioned strengths that make modulation mechanism suitable for efficient networks, we also tentatively introduce a novel perspective in Appendix Sec. K that modulation has the unique potential to project the input feature to a very high dimensional space.

## 3.2 EFFICIENT MODULATION

Despite being more efficient than self-attention, the abstracted modulation mechanism still fails to meet the efficiency requirements of mobile networks in terms of theoretical complexity and inference latency. Here, we introduce Efficient Modulation, which is tailored for efficient networks but retains all the desirable properties of the modulation mechanism.

**Sliming Modulation Design.** A general modulation block has many fragmented operations, as illustrated in Fig. 1b. Four FC layers are introduced without considering the details of the context modeling implementation. As stated in guideline G3 in ShuffleNetv2 (Ma et al., 2018), too many fragmented operations will significantly reduce speed, even if the computational complexity may be low by tweaking the channel number. To this end, we fuse the FC layers from the MLP and modulation blocks as shown in Fig.1c. We consider $v\left(\cdot\right)$ to expand the channel dimension by an expansion factor of $r$ and leverage $p\left(\cdot\right)$ to squeeze the channel number. That is, the MLP block is fused into our modulation design with a flexible expansion factor, resulting in a unified block similar to the MBConv block (we will discuss the differences and show our superiority in Table. 6).

**Simplifying Context Modeling.** We next tailor our context modeling branch for efficiency. given the input $x$, we first project $x$ to a new feature space by a linear projection $f\left(x\right)$. Then, a depth-wise convolution with GELU activation is employed to model local spatial information. We set the kernel size to 7 to balance the trade-off between efficiency and a large receptive field. Lastly, a linear projection $g\left(x\right)$ is employed for channel communication. Notice that the channel number is kept the same throughout the context modeling branch. In short, our context modeling branch can be given by:

$$\text{ctx}\left(x\right) = g\left(\text{act}\left(\text{DWConv}_{7,1}\left(f\left(x\right)\right)\right)\right). \tag{5}$$

This design is much simpler than the context modeling in VAN and FocalNet. We discard isofunctional depth-wise convolutions by one large-kernel depth-wise convolution. We acknowledge that this may slightly degrade the performance as a compromise to efficiency. Ablation studies demonstrate that each operation in our context branch is indispensable.

## 3.3 NETWORK ARCHITECTURE

With the modifications as mentioned above, we arrive at our Efficient Modulation block depicted in Fig. 1c. Next, we instantiate our efficient networks. Please see Appendix Sec. B for more details.

First, we introduce a pure convolutional network solely based on the EfficientMod block. Following common practice (Li et al., 2023b; Yu et al., 2022a), we adopt a hierarchical architecture of 4 stages; each stage consists of a series of our EfficientMod blocks with residual connection. For simplicity, we used overlapped patch embedding (implemented with a convolutional layer) to down-size the features by a factor of 4, 2, 2, and 2, respectively. For each block, we normalize the input feature using Layer Normalization (Ba et al., 2016) and feed the normalized feature to our EfficientMod block. We employ Stochastic Depth (Huang et al., 2016) and Layer Scale (Touvron et al., 2021b) to improve the robustness of our model. Notice that our EfficientMod block is orthogonal to the self-attention mechanism. Following recent advances that combine convolution and attention for better performance (Li et al., 2023b; Mehta & Rastegari, 2022; Chen et al., 2022b; Pan et al., 2022), we next combine our EfficientMod with attention block to get a new hybrid design. We consider the vanilla attention block as in ViT (Dosovitskiy et al., 2021) without any modifications. The attention blocks are only introduced in the last two stages, where the feature size is relatively small. We vary the width and depth to match the parameters in the pure convolutional-based EfficientMod counterpart for a fair comparison. We introduce three scales ranging from 4M to 13M parameters, resulting in EfficientMod-xxs, EfficientMod-xs, and EfficientMod-s.

### 3.4 Computational Complexity analysis

We also examine our design's theoretical computational complexity and practical guidelines.

Given input feature $x \in \mathbb{R}^C \times H \times W$, the total parameters number of one EfficientMod block is $2(r+1)C^2 + k^2C$, and the computational complexity is $\mathcal{O}\left(2(r+1)HWC^2 + HWk^2C\right)$, where $k$ is kernel size and $r$ is the expansion ratio in $v(\cdot)$. We ignore the activation function and bias in learnable layers for simplicity. Compared with Attention, our complexity is linear to the input resolution. Compared with MBConv, we reduce the complexity of depth-wise convolution by a factor of $r$, which is crucial for effectiveness as validated in Table 6.

Besides the theoretical computational complexity, we also provide some practical guidelines for our design. **I)** *We reduce the FLOPs by moving more parameters to later stages where the feature resolution is small.* The reason behind is that our EfficientMod's FLOPs are basically equal to *the input resolution × the number of parameters*. Following this guideline, we can add more blocks or substantially increase the width in later stages. Note that this guideline is not unique to our EfficientMod and can be applied to all FC and Convolutional layers. **II)** *We only introduce attention blocks to the last two stages*, as a common practice in many works (Li et al., 2023b; Mehta & Rastegari, 2022; Yu et al., 2022b; Mehta & Rastegari, 2023) considering self-attention's computational complexity. **III)** *We use* Repeat *operation to match channel number to save CPU time with a light overhead on GPU.* EfficientFormer observed that the Reshape

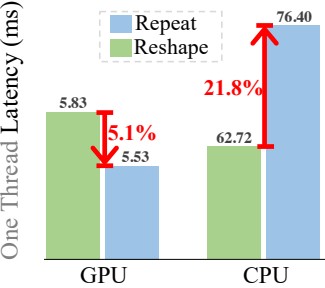

Figure 2: From Repeat to Reshape, EfficientMod-s GPU latency decreases 5.1% and CPU latency increases 21.8%.

is often a bottleneck for many models. Here, we introduce more details. Reshape is considerably sluggish on the CPU but is GPU-friendly. Meanwhile, Repeat operation is swift on CPU but time-consuming on GPU. As shown in Fig. 2, two solutions (Repeat and Reshape) can be used for interact in EfficientMod, we select Repeat to get the optimal GPU-CPU latency trade-off.

### 3.5 Relation to Other Models

Lastly, we discuss the connections and differences between our EfficientMod block and other notable designs to emphasize the unique properties of our approach.

**MobileNetV2** ushered in a new era in the field of efficient networks by introducing mobile inverted bottleneck (MBConv in short) block. Compared to the MBConv block that sequentially arranges the FC layer, our EfficientMod block separates the depth-wise convolutional layer and inserts it from the side into the middle of the two-layer FC network via element-wise multiplication. We will show that our design is a more efficient operation (due to the channel number reduction of depth-wise convolution) and achieve better performance (due to modulation operation) in Table 6.

Table 1: **ImageNet-1K classification performance.** We compare EfficientMod with SOTA methods and report inference latency, model parameters, and FLOPs. The latency is measured on one P100 GPU and Intel(R) Xeon(R) CPU E5-2680 CPU with four threads. We use tiny gray color to indicate results trained with strong training strategies like re-parameterization in MobileOne and distillation in EfficientFormerV2. Benchmark results on more GPUs can be found in Appendix Sec. H.

| Model | Top-1(%) | Latency (ms) | | Params (M) | FLOPs (G) | Size. |
|---|---|---|---|---|---|---|
| | | GPU | CPU | | | |
| MobileNetV2×1.0 (2018) | 71.8 | 2.1 | 3.8 | 3.5 | 0.3 | $224^2$ |
| FasterNet-T0 (2023) | 71.9 | 2.5 | 6.8 | 3.9 | 0.3 | $224^2$ |
| EdgeViT-XXS (2022) | 74.4 | 8.8 | 15.7 | 4.1 | 0.6 | $224^2$ |
| MobileOne-S1 (2023) | 74.6 (75.9) | 1.5 | 6.9 | 4.8 | 0.8 | $224^2$ |
| MobileViT-XS (2022) | 74.8 | 4.1 | 21.0 | 2.3 | 1.1 | $256^2$ |
| EfficientFormerV2-S0 (2023b) | 73.7 (75.7) | 3.3 | 10.7 | 3.6 | 0.4 | $224^2$ |
| EfficientMod-xxs | **76.0** | 3.0 | 10.2 | 4.7 | 0.6 | $224^2$ |
| MobileNetV2×1.4 (2018) | 74.7 | 2.8 | 6.0 | 6.1 | 0.6 | $224^2$ |
| DeiT-T (2021a) | 74.5 | 2.7 | 16.5 | 5.9 | 1.2 | $224^2$ |
| FasterNet-T1 (2023) | 76.2 | 3.3 | 12.9 | 7.6 | 0.9 | $224^2$ |
| EfficientNet-B0 (2019) | 77.1 | 3.4 | 10.9 | 5.3 | 0.4 | $224^2$ |
| MobileOne-S2 (2023) | - (77.4) | 2.0 | 10.0 | 7.8 | 1.3 | $224^2$ |
| EdgeViT-XS (2022) | 77.5 | 11.8 | 21.4 | 6.8 | 1.1 | $224^2$ |
| MobileViTv2-1.0 (2023) | 78.1 | 5.4 | 30.9 | 4.9 | 1.8 | $256^2$ |
| EfficientFormerV2-S1 (2023b) | 77.9 (79.0) | 4.5 | 15.4 | 6.2 | 0.7 | $224^2$ |
| EfficientMod-xs | **78.3** | 3.6 | 13.4 | 6.6 | 0.8 | $224^2$ |
| PoolFormer-s12 (2022a) | 77.2 | 5.0 | 22.3 | 11.9 | 1.8 | $224^2$ |
| FasterNet-T2 (2023) | 78.9 | 4.4 | 18.4 | 15.0 | 1.9 | $224^2$ |
| EfficientFormer-L1 (2022) | 79.2 | 3.7 | 19.7 | 12.3 | 1.3 | $224^2$ |
| MobileFormer-508M (2022b) | 79.3 | 13.4 | 142.5 | 14.8 | 0.6 | $224^2$ |
| MobileOne-S4▲ (2023) | - (79.4) | 4.8 | 26.6 | 14.8 | 3.0 | $224^2$ |
| MobileViTv2-1.5 (2023) | 80.4 | 7.2 | 59.0 | 10.6 | 4.1 | $256^2$ |
| EdgeViT-S (2022) | **81.0** | 20.5 | 34.7 | 13.1 | 1.9 | $224^2$ |
| EfficientFormerV2-S2 (2023b) | 80.4 (81.6) | 7.3 | 26.5 | 12.7 | 1.3 | $224^2$ |
| EfficientMod-s | **81.0** | 5.5 | 23.5 | 12.9 | 1.4 | $224^2$ |

**SENet** introduces dynamics to ConvNets by proposing channel-attention mechanism (Hu et al., 2018). An SE block can be given by $y = x \cdot \mathrm{sig}\left(W_2\left(\mathrm{act}\left(W_1 x\right)\right)\right)$. Many recent works (Tan & Le, 2019; Zhang et al., 2022; Liu et al., 2022a) incorporate it to achieve better accuracy while maintaining a low complexity in theory. However, due to the fragmentary operations in SE block, it would significantly reduce the inference latency on GPUs. On the contrary, our EfficientMod block inherently involves channel attention via $y = \mathrm{ctx}\left(x\right) \cdot \mathrm{q}\left(x\right)$, where $\mathrm{q}\left(x\right)$ adaptively adjust the channel weights of $\mathrm{ctx}\left(x\right)$.

## 4 EXPERIMENTS

In this section, we validate our EfficientMod on four tasks: image classification on ImageNet-1K (Deng et al., 2009), object detection and instance segmentation on MS COCO (Lin et al., 2014), and semantic segmentation on ADE20K (Zhou et al., 2017). We implement all networks in PyTorch and convert to ONNX models on two different hardware:

- **GPU:** We chose the P100 GPU for our latency evaluation since it can imitate the computing power of the majority of devices in recent years. Other GPUs may produce different benchmark results, but we observed that the tendency is similar.
- **CPU:** Some models may operate with unpredictable latency on different types of hardware (mostly caused by memory accesses and fragmented operations). We also provide all models' measured latency on the Intel(R) Xeon(R) CPU E5-2680 CPU for a full comparison.

For the latency benchmark, we set the batch size to 1 for both GPU and CPU to simulate real-world applications. To counteract the variance, we repeat 4000 runs for each model and report the mean inference time. We use four threads following the common practice. For details on more devices (*e.g.*, different GPUs, iPhone, *etc.*), please check out supplemental material.

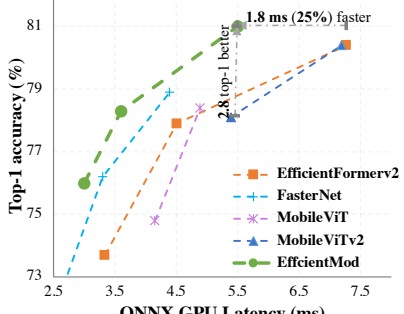

Figure 3: The trade-off between ONNX GPU latency and accuracy.

| Arch. | Model | Params | FLOPs | Acc. | Epoch |
|---|---|---|---|---|---|
| | RSB-ResNet-18 (2021) | 12.0M | 1.8G | 70.6 | 300 |
| | RepVGG-A1 (2021) | 12.8M | 2.4G | 74.5 | 120 |
| | PoolFormer-s12 (2022a) | 11.9M | 1.8G | 77.2 | 300 |
| | GhostNetv2×1.6 (2022) | 12.3M | 0.4G | 77.8 | 450 |
| Conv | RegNetX-3.2GF (2020) | 15.3M | 3.2G | 78.3 | 100 |
| | FasterNet-T2 (2023) | 15.0M | 1.9G | 78.9 | 300 |
| | ConvMLP-M (2023a) | 17.4M | 3.9G | 79.0 | 300 |
| | GhostNet-A (2020b) | 11.9M | 0.6G | 79.4 | 450 |
| | MobileOne-S4 (2023) | 14.8M | 3.0G | 79.4 | 300 |
| | EfficientMod-s | 12.9M | 1.5G | **80.5** | 300 |
| + Atten. | EfficientMod-s | 12.9M | 1.4G | **81.0** | 300 |

Table 2: We compare our convolution-based model with others and show improvements in the hybrid version.

## 4.1 IMAGE CLASSIFICATION ON IMAGENET-1K

We evaluate the classification performance of EfficientMod networks on ImageNet-1K. Our training recipe follows the standard practice in DeiT (Touvron et al., 2021a), details can be found in Appendix Sec. 5. Strong training tricks (*e.g.*, re-parameterization and distillation) were not used to conduct a fair comparison and guarantee that all performance was derived from our EfficientMod design.

We compare our EfficientMod with other efficient designs and present the results in Table 1. Distinctly, our method exhibits admirable performance in terms of both classification accuracy and inference latency on different hardware. For instance, our EfficientMod-s performs the same as EdgeViT but runs 15 milliseconds (about 73%) faster on the GPU and 11 milliseconds (about 32%) faster on the CPU. Moreover, our model requires fewer parameters and more minor computational complexity. EfficientMod-s also outperforms EfficientFormerV2-S2 by **0.6 improvements and runs 1.8ms (about 25%) faster on GPU**. Our method performs excellently for different scales. Be aware that some efficient designs (like MobileNetV2 and FasterNet) prioritize low latency while other models prioritize performance (like MobileViTv2 and EdgeViT). In contrast, our EfficientMod provides state-of-the-art performance while running consistently fast on both GPU and CPU.

To better grasp the enhancements of our method, we use EfficientMod-s as an example and outline the specific improvements of each modification. The results of our EfficientMod, from the pure convolutional-based version to the hybrid model, are presented in Table 2.

We note that even the pure convolutional-based version of EfficientMod already produces impressive results at 80.5%, significantly surpassing related convolutional-based networks. By adapting to hybrid architecture, we further enhance the performance to 81.0%.

| EFormerv2 | s0 (3.3ms) | s1 (4.5ms) | s2 (7.3ms) |
|---|---|---|---|
| w/o Distill. | 73.7 | 77.9 | 80.4 |
| w/ Distill. | 75.7 (+2.0) | 79.0 (+1.1) | 81.6 (+1.2) |
| EfficientMod | xxs (3.0ms) | xs (3.6ms) | s (5.5ms) |
| w/o Distill. | 76.0 | 78.3 | 81.0 |
| w/ Distill. | 77.1 (+1.1) | 79.4 (+1.1) | 81.9 (+0.9) |

Table 3: Results w/o and w/ distillation.

Meanwhile, some methods are trained with strong training strategies, like re-parameterization (Ding et al., 2021) in MobileOne and distillation (Hinton et al., 2015) in EfficientFormerV2. When trained with distillation (following the setting in (Li et al., 2023b)), we improve EfficientMod-s from 81.0 to 81.9%, as shown in Table 3. All following results are without distillation unless stated otherwise.

## 4.2 ABLATION STUDIES

**Compare to other Modulation models.** We compare our EfficientMod-xxs with FocalNet and VAN-B0, that has a similar number of parameters. For a fair comparison, we customize Focal-

| Model | Top-1(%)↑ | GPU (ms)↓ | CPU (ms)↓ | Param. | FLOPs |
|---|---|---|---|---|---|
| VAN-B0 | 75.4 (↓0.6) | 4.5 (↑1.5) | 16.3 (↑6.1) | 4.1M | 0.9G |
| FocalNet@4M | 74.5 (↓1.5) | 4.2 (↑1.2) | 16.8 (↑6.6) | 4.6M | 0.7G |
| EfficientMod-xxs | **76.0** | **3.0** | **10.2** | 4.7M | 0.6G |

Table 4: Compare EfficientMod with other modulation models.

Net_Tiny_lrf by reducing the channel number or the blocks. We tested three variants, selected the best one, and termed it FocalNet@4M. Since Conv2Former (Hou et al., 2022) code has not been fully released, we didn't consider it in our comparison. From Table 4, we see that EfficientMod outperforms other modulation methods for both accuracy and latency.

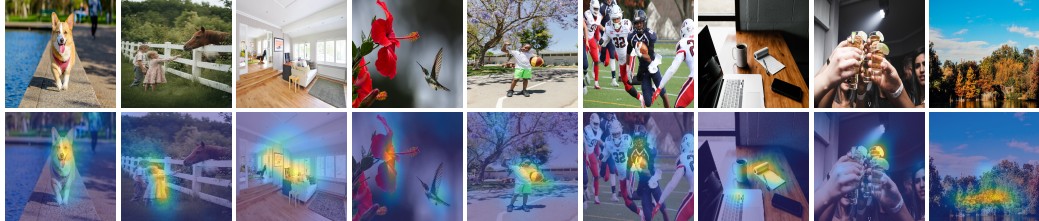

Figure 4: We directly visualize the forward context modeling results as shown in Eq. 5. The visualization results suggest that our context modeling can emphasize the conspicuous context. No backward gradient is required as in Class Activation Map (Zhou et al., 2016).

| $f(\cdot)$ | Conv | $g(\cdot)$ | Acc. |
|---|---|---|---|
| ✓ | | | 72.7 -7.8 |
| | ✓ | | 78.6 -0.9 |
| | | ✓ | 72.3 -8.2 |
| ✓ | ✓ | | 79.8 -0.7 |
| | ✓ | ✓ | 79.6 -0.9 |
| ✓ | ✓ | ✓ | **80.5** |
| mul. → sum | | | 79.5 -1.0 |

Table 5: Ablation studies based on EfficientMod-s-Conv w/o attention.

| Arch. | Model | Params | FLOPs | Acc. | Latency (ms) GPU | Latency (ms) CPU |
|---|---|---|---|---|---|---|
| iso. | MBConv | 6.4M | 1.6G | **72.9** | 4.4 | 122.2 |
| | EfficientMod | 6.4M | 1.6G | **72.9** | **2.9** | **19.3** |
| | MBConv | 12.4M | 3.1G | 77.0 | 6.5 | 196.5 |
| | EfficientMod | 12.5M | 3.1G | **77.6** | **4.2** | **39.1** |
| hier. | MBConv | 6.4M | 0.7G | 76.9 | 5.4 | 18.7 |
| | EfficientMod | 6.4M | 0.7G | **77.4** | **3.8** | **12.4** |
| | MBConv | 12.9M | 1.6G | 79.8 | 9.2 | 59.6 |
| | EfficientMod | 12.9M | 1.5G | **80.5** | **5.8** | **25.0** |

Table 6: Comparison between MBConv and EfficientMod with isotropic (iso.) and hierarchical (hier.) architecture.

**Ablation of each component.** We start by examining the contributions provided by each component of our design. Experiments are conducted on the convolutional EfficientMod-s without introducing attention and knowledge distillation. Table 5 shows the results of eliminating each component in the context modeling branch. Clearly, all these components are critical to our final results. Introducing all, we arrive at 80.5% top-1 accuracy. Meanwhile, we also conducted an experiment to validate the effectiveness of element-wise multiplication. We substitute it with summation (same computations and same latency) to fuse features from two branches and present the results in the last row of the table. As expected, the performance drops by 1% top-1 accuracy. The considerable performance drop reveals the effectiveness of our modulation operation, especially in efficient networks.

**Connection to MBConv blocks.** To verify the superiority of EfficientMod block, we compare our design and the essential MBConv with isotropic and hierarchical architectures, respectively. Please check Appendix Sec. B for detailed settings. With almost the same number of parameters and FLOPs, results in Table 6 indicate that our EfficientMod consistently runs faster than MBConv counterparts by a significant margin on both GPU and CPU. One most probable explanation is that our depth-wise convolution is substantially lighter than MBConv's (channel numbers are $c$ and $rc$, respectively, where $r$ is set to 6). Besides the faster inference, our design consistently provides superior empirical results than MBConv block. Please check Appendix Sec. G for more studies on scalability.

**Context Visualization.** Inherited from modulation mechanism, our EfficientMod block can distinguish informative context. Following FocalNet, we visualize the forward output of the context layer (computing the mean value along channel dimension) in EfficientMod-Conv-s, as shown in Fig. 4. Clearly, our model consistently captures the informative objects, and the background is restrained, suggesting the effectiveness of the modulation design in efficient networks.

## 4.3 OBJECT DETECTION AND INSTANCE SEGMENTATION ON MS COCO

To validate the performance of EfficientMod on downstream tasks, we conduct experiments on MS COCO dataset for object detection and instance segmentation. We validate our EfficientMod-s on top of the common-used detector Mask RCNN (He et al., 2017). We follow the implementation of previous work (Yu et al., 2022a; Wang et al., 2021; 2022; Tan & Le, 2021), and train the model using 1× scheduler, *i.e.*, 12 epochs. We compare our convolutional and hybrid EfficientMod-s with other methods and report the results in Table 7. Results suggest that EfficientMod consistently outper-

Table 7: Performance in downstream tasks. We equip all backbones with Mask-RCNN and train the model with (1×) scheduler for detection and instance segmentation on MS COCO. We consider Semantic FPN for semantic segmentation on ADE20K. Our pre-trained weights are from Table 3.

| Arch. | Backbone | MS COCO | | | | | | | ADE20K | | |
|---|---|---|---|---|---|---|---|---|---|---|---|
| | | Params | $AP^b$ | $AP^b_{50}$ | $AP^b_{75}$ | $AP^m$ | $AP^m_{50}$ | $AP^m_{75}$ | Params | FLOPs | mIoU |
| Conv. | ResNet-18 | 31.2M | 34.0 | 54.0 | 36.7 | 31.2 | 51.0 | 32.7 | 15.5M | 32.2G | 32.9 |
| Pool | PoolF.-S12 | 31.6M | 37.3 | 59.0 | 40.1 | 34.6 | 55.8 | 36.9 | 15.7M | 31.0G | 37.2 |
| Conv. | EfficientMod-s | 32.6M | **42.1** | **63.6** | **45.9** | **38.5** | **60.8** | **41.2** | 16.7M | 29.0G | **43.5** |
| Atten. | PVT-Tiny | 32.9M | 36.7 | 59.2 | 39.3 | 35.1 | 56.7 | 37.3 | 17.0M | 33.2G | 35.7 |
| Hybrid | EfficientF.-L1 | 31.5M | 37.9 | 60.3 | 41.0 | 35.4 | 57.3 | 37.3 | 15.6M | 28.2G | 38.9 |
| Hybrid | PVTv2-B1 | 33.7M | 41.8 | 64.3 | 45.9 | 38.8 | 61.2 | 41.6 | 17.8M | 34.2G | 42.5 |
| Hybrid | EfficientF.v2-s2 | 32.2M | 43.4 | 65.4 | 47.5 | 39.5 | 62.4 | 42.2 | 16.3M | 27.7G | 42.4 |
| Hybrid | EfficientMod-s | 32.6M | **43.6** | **66.1** | **47.8** | **40.3** | **63.0** | **43.5** | 16.7M | 28.1G | **46.0** |

forms other methods with similar parameters. Without self-attention, our EfficientMod surpasses PoolFormer by 4.2 mAP for detection and 3.6 mAP on instance segmentation task. When introducing attention and compared with hybrid models, our method still outperforms others on both tasks.

## 4.4 Semantic Segmentation on ADE20K

We next conduct experiments on the ADE20K (Zhou et al., 2017) dataset for the semantic segmentation task. We consider Semantic FPN (Kirillov et al., 2019) as the segmentation head due to its simple and efficient design. Following previous work (Yu et al., 2022a; Li et al., 2023b; 2022; Wang et al., 2021), we train our model for 40k iterations with a total batch size of 32 on 8 A100 GPUs. We train our model using AdamW (Loshchilov & Hutter, 2019) optimizer. The Cosine Annealing scheduler (Loshchilov & Hutter, 2017) is used to decay the learning rate from initialized value 2e-4.

Results in Table 7 demonstrate that EfficientMod outperforms other methods by a substantial margin. Without the aid of attention, **our convolutional EfficientMod-s already outperforms PoolFormer by 6.3 mIoU**. Furthermore, the pure convolutional EfficientMod even achieves better results than the attention-equipped methods. In this regard, our convolutional EfficientMod-s performs 1.1 mIoU better than the prior SOTA efficient method EfficientFormerV2 (42.4 *vs.* 43.5). *The design of our EfficientMod block is the sole source of these pleasing improvements*. When introducing Transformer blocks to get the hybrid design, we further push the performance to 46.0 mIoU, using the same number of parameters and even fewer FLOPs. Hybrid EfficientMod-s performs noticeably better than other hybrid networks, **outperforming PvTv2 and EfficientFormerV2 by 3.5 and 3.6 mIoU, respectively**. Two conclusions are offered: 1) EfficientMod design makes significant advancements, demonstrating the value and effectiveness of our approach; 2) Large receptive fields are especially helpful for high-resolution input tasks like segmentation, and the vanilla attention block (which achieves global range) can be an off-the-shelf module for efficient networks. Please check Appendix Sec. F for the analysis of the improvement gap between MS COCO and ADE20K.

## 5 Conclusion

We present Efficient Modulation (EfficientMod), a unified convolutional-based building block that incorporates favorable properties from both convolution and attention mechanisms. EfficientMod simultaneously extracts the spatial context and projects input features, and then fuses them using a simple element-wise multiplication. EfficientMod's elegant design gratifies efficiency, while the inherent design philosophy guarantees great representational ability. With EfficientMod, we built a series of efficient models. Extensive experiments examined the efficiency and effectiveness of our method. EfficientMod outperforms previous SOTA methods in terms of both empirical results and practical latency. When applied to dense prediction tasks, EfficientMod delivered impressive results. Comprehensive studies indicate that our method has great promise for efficient applications.

**Limitations and Broader Impacts.** The scalability of efficient designs is one intriguing but understudied topic, like the huge latency gap in Table 6. Also, employing large kernel sizes or introducing attention blocks might not be the most efficient way to enlarge the receptive field. We have not yet observed any negative societal impacts from EfficientMod. Instead, we encourage study into reducing computations and simplifying real-world applications with limited computational resources.

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

## A  CODES AND MODELS

The codes can be found in the supplemental material. We provide anonymous links to the pre-trained checkpoints and logs. The ReadME.md file contains thorough instructions to conduct experiments. After the submission, we will make our codes and pre-trained checkpoints available.

## B  DETAILED CONFIGURATIONS

| Stage | size | EfficientMod-xxs | EfficientMod-xs | EfficientMod-s | EfficientMod-s$_{(Conv)}$ |
|---|---|---|---|---|---|
| Stem | $\frac{H}{4} \times \frac{W}{4}$ | Conv(kernel=7, stride=4) | | | |
| Stage 1 | $\frac{H}{4} \times \frac{W}{4}$ | Dim=32 Blocks = $[2,0]$ | Dim=32 Blocks = $[3,0]$ | Dim=32 Blocks = $[4,0]$ | Dim=40 Blocks = $[4,0]$ |
| Down | $\frac{H}{8} \times \frac{W}{8}$ | Conv(kernel=3, stride=2) | | | |
| Stage 2 | $\frac{H}{8} \times \frac{W}{8}$ | Dim=64 Blocks = $[2,0]$ | Dim=64 Blocks = $[3,0]$ | Dim=64 Blocks = $[4,0]$ | Dim=80 Blocks = $[4,0]$ |
| Down | $\frac{H}{16} \times \frac{W}{16}$ | Conv(kernel=3, stride=2) | | | |
| Stage 3 | $\frac{H}{16} \times \frac{W}{16}$ | Dim=128 Blocks = $[6,1]$ | Dim=144 Blocks = $[4,3]$ | Dim=144 Blocks = $[8,4]$ | Dim=160 Blocks = $[12,0]$ |
| Down | $\frac{H}{32} \times \frac{W}{32}$ | Conv(kernel=3, stride=2) | | | |
| Stage 4 | $\frac{H}{32} \times \frac{W}{32}$ | Dim=256 Blocks = $[2,2]$ | Dim=288 Blocks = $[2,3]$ | Dim=312 Blocks = $[8,4]$ | Dim=344 Blocks = $[8,0]$ |
| Head | $1 \times 1$ | Global Average Pooling & MLP | | | |
| Parameters (M) | | 4.7 | 6.6 | 12.9 | 12.9 |

Table 8: Detailed configuration of our EfficientMod architecture. Dim denotes the input channel number for each stage. Blocks $[b_1, b_2]$ indicates we use $b_1$ EfficientMod blocks and $b_2$ vanilla attention blocks, respectively. For our EfficientMod block, we alternately expand the dimension by a factor of 1 and 6 (1 and 4 for EfficientMod-xs). For the vanilla attention block, we consider 8 heads by default.

**Detailed Framework Configurations** We provide a detailed configuration of our EfficientMod in Table 8. Our EfficientMod is a hierarchical architecture that progressively downsizes the input resolution by $4, 2, 2, 2$ using the traditional convolutional layer. For the stages that include both EfficientMod and attention blocks, we employ EfficientMod blocks first, then utilize the attention blocks. By varying channel and block numbers, we introduce EfficientMod-xxs, EfficientMod-xs, EfficientMod-s, and a pure convolutional version of EfficientMod-s.

**Detailed Ablation Configurations** For the isotropic designs in Table 6, we patchify the input image using a $14 \times 14$ patch size, bringing us a resolution of $16 \times 16$. We adjust the depth and width to deliberately match the number of parameters as EfficientMod-xs and EfficientMod-s. We also vary the expansion ratio to match the number of parameters and FLOPs for MBConv and EfficientMod counterparts. We chose 256 and 196 for the channel number and 13 and 11 for the depth, respectively. Similar to EfficientMod-s and EfficientMod-xs, the generated models will have 12.5M and 6.4M parameters, respectively. By doing so, we guarantee that any performance differences result purely from the design of the MBConv and EfficientMod blocks. For hierarchical networks, we replace the EfficientMod block with the MBConv block and vary the expansion ratio to the match parameter number.

| Hyper parameters | EfficientMod |
|---|---|
| Batch size | 256×8 = 2048 |
| Optimizer | AdamW |
| Weight decay | 0.05 |
| Clip-grad | None |
| LR scheduler | Cosine |
| Learning rate | 4e-3 |
| Epochs | 300 |
| Warmup epochs | 5 |
| Hflip | 0.5 |
| Vflip | 0. |
| Color-jitter | 0.4 |
| AutoAugment | rand-m9-mstd0.5-inc1 |
| Aug-repeats | 0 |
| Random erasing prob | 0.25 |
| Mixup | 0.8 |
| Cutmix | 1.0 |
| Label smoothing | 0.1 |
| Layer Scale | 1e-4 |
| Drop path | {0., 0., 0.02} |
| Drop block | 0. |

Figure 5: Training hyper-parameters.

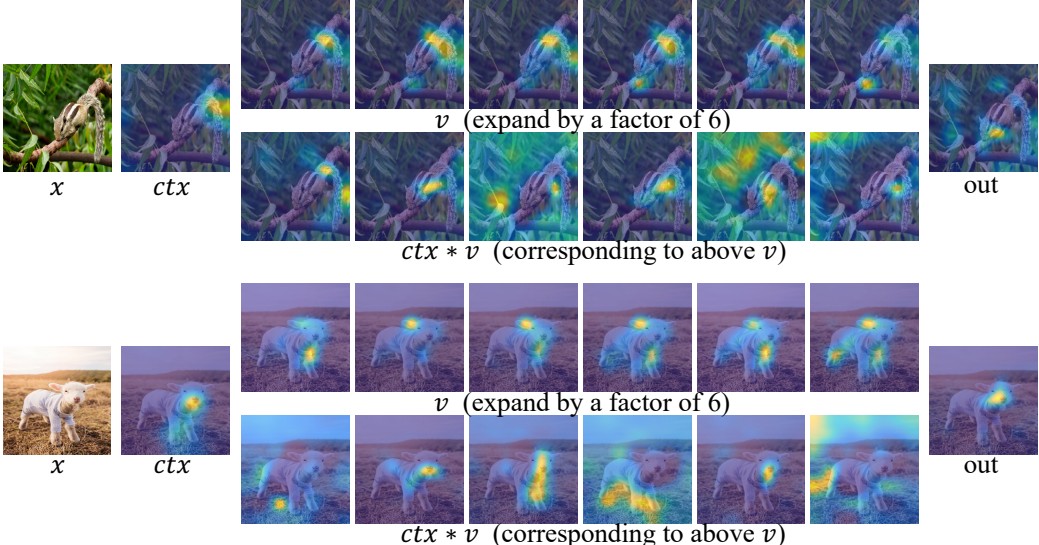

Figure 6: Visualization of each expanded $v$ and associated modulation result $ctx * v$ in more detail. We provide the final output and the context modeling result ($ctx$) for reference.

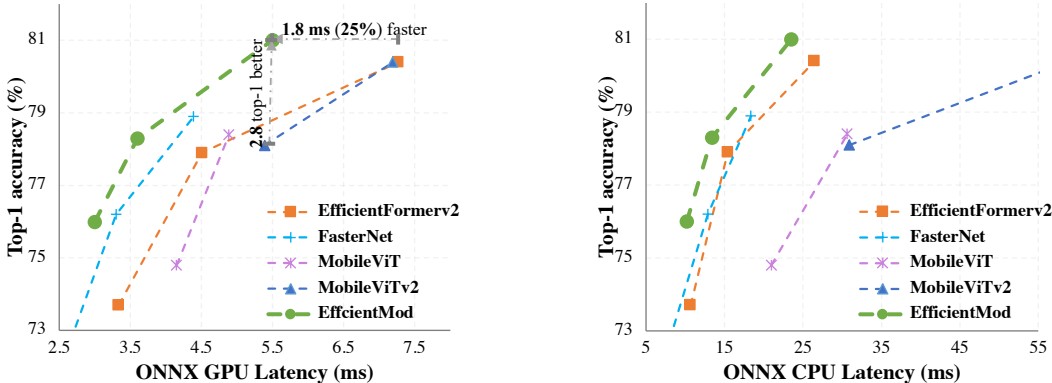

Figure 7: Comparison of the Latency-Accuracy trade-off between EfficientMod and other methods on GPU and CPU devices. EfficientMod consistently performs far better than other methods.

**Training details** The detailed training hyper parameters are presented in Table 5.

## C  VISUALIZATION OF MODULATION

We provide a complete visualization of our modulation design, as seen in Fig.6. We showcase the project input $v$, the context modeling output $ctx$, and the corresponding modulation outcomes $ctx * v$. The block's output is presented last. The visualization implementation is the same as the settings in Section 4.2. We divide the feature into $r$ chunks along the channel dimension when the input feature is expanded by a factor of $r$ (for example, 6 in Fig. 6), and we next visualize each chunk and its accompanying modulation result. Interestingly, as the channel count increases, different thunks show generally similar results $v$. The difference is substantially accentuated after being modulated by the same context, indicating the success of the modulation mechanism.

## D  LATENCY COMPARISON

We also demonstrate the accuracy versus GPU and CPU latency for various methods, as seen in Fig. 7. We eliminate specific models that run significantly slower (such as EdgeViT on GPU) or yield

much lower accuracy (such as MobileNetV2) for better visualization. Our approach consistently surpasses related work, especially on GPU, by a clear margin. Our EfficientMod outperforms MobileViTv2 by 2.8 top-1 accuracy on ImageNet with the same GPU latency. We are 25% faster than the previous state-of-the-art approach EffcientFormerv2 to obtain results that are similar or slightly better (0.3%). On the CPU, we also achieve a promising latency-accuracy trade-off, demonstrating that our EfficientMod can be utilized as a general method on different devices.

## E    DISTILLATION IMPROVEMENTS

We provide detailed experiments for distillation on each model. Our teacher model is the widely used RegNetY-160 (Radosavovic et al., 2020), the same as the teacher in EfficientFormerV2 for fair comparison. Though other models may offer superior improvements, a better teacher model is not the primary objective of this study. The right table shows that knowledge distillation is a potent way to improve our method's top-1 accuracy about 1% without introducing any computational overhead during inference.

| Model | Param | FLOPs | Distill. | Top-1 |
|---|---|---|---|---|
| EfficientMod-xxs | 4.7M | 0.6G | ✗ | 76.0 |
| | | | ✓ | 77.1 (↑1.1) |
| EfficientMod-xs | 6.6M | 0.8G | ✗ | 78.3 |
| | | | ✓ | 79.4 (↑1.1) |
| EfficientMod-s | 12.9M | 1.4G | ✗ | 81.0 |
| | | | ✓ | 81.9 (↑0.9) |
| EfficientMod-s (Conv) | 12.9M | 1.5G | ✗ | 80.5 |
| | | | ✓ | 81.5 (↑1.0) |

Table 9: Detailed improvements from Distillation.

## F    ANALYSIS ON IMPROVEMENT GAP BETWEEN OBJECT DETECTION AND SEMANTIC SEGMENTATION

*Why does ModelMod improve significantly on ADE20K but only modestly on MS COCO?* Besides the differences in datasets and evaluation metrics, *we attribute this discrepancy to the number of parameters in detection or segmentation head*. Notice that EfficientMod only introduces 12M parameters. When equipped with Mask RCNN, the additional parameters are over 20M ( over 60% in total), dominating the final detection network. Hence, the impact of the backbone is largely inhibited. On the contrary, Semantic FPN only introduces 4-5M parameters for semantic segmentation on ADE20K (about 25% in total). Hence, EfficientMod's capabilities are fully utilized.

## G    SCALABILITY OF EFFICIENTMOD

**Latency against input resolution**. We first validate our scalability for the input resolution. We vary the input resolution from 224 to 512 by a step size of 32. We compare our convolutional and hybrid variants of EfficientMod with some strong baselines, including MobileFormer (Chen et al., 2022b), MobileViTv2 (Mehta & Rastegari, 2023), and EfficientFormerV2 (Li et al., 2023b). For a fair comparison, we consider model size in the range of 10-15M parameters for all models. From Fig. 8, we can observe that our method and EfficientFormerV2 show promising scalability to the input resolution when compared with MobileFormer and MobileViTv2. Compared to EfficientFormerV2, our method (both convolutional and hybrid) also exhibits even lower latency when the resolution is small.

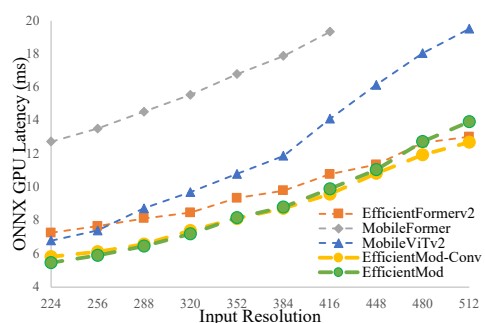

Figure 8: Impact of input size on GPU latency.

**Compare with MBConv**. We also investigate the scalability of the width and kernel size for our EfficientMod. We compare EfficientMod and the commonly used MBConv from MobileNetV2 (Sandler et al., 2018) using the same settings described in Sec. B. To match the computational complexity and parameter number, the expansion ratio is set to 6 and 7 for EfficientMod and MBConv, respectively. As shown in Fig. 9, our EfficientMod consistently runs faster than MBConv block regardless of width

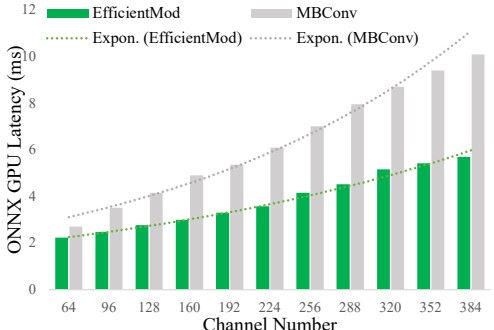 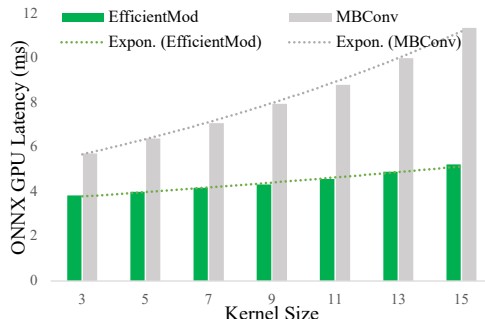

Figure 9: Scalability comparison between our EfficientMod and MBConv blocks by varying the width and kernel size. We use a dotted line to show the latency tendency. Model architecture is inherited from Table 6 isotropic design.

Table 10: Latency benchmark results across multiple GPU instances. We conducted three tests for each GPU on different nodes and averaged the results from 500 runs for each test. We report the mean$_{\pm std}$ latency in the table. The setting is the same as in Table 1.

| Model | Top-1(%) | **P100** | **T4** | **V100-SXM2** | **A100** | Param. | FLOPs |
|---|---|---|---|---|---|---|---|
| MobileNetV2×1.0 | 71.8 | 2.2$_{\pm0.003}$ | 1.4$_{\pm0.000}$ | 1.1$_{\pm0.003}$ | 1.3$_{\pm0.003}$ | 3.5M | 0.3G |
| FasterNet-T0 | 71.9 | 2.5$_{\pm0.000}$ | 1.8$_{\pm0.010}$ | 1.7$_{\pm0.013}$ | 2.0$_{\pm0.063}$ | 3.9M | 0.3G |
| EdgeViT-XXS | 74.4 | 8.8$_{\pm0.000}$ | 4.7$_{\pm0.023}$ | 2.4$_{\pm0.000}$ | 2.7$_{\pm0.003}$ | 4.1M | 0.6G |
| MobileViT-XS | 74.8 | 4.2$_{\pm0.003}$ | 3.5$_{\pm0.000}$ | 2.3$_{\pm0.003}$ | 2.4$_{\pm0.000}$ | 2.3M | 1.1G |
| EfficientFormerV2-S0 | 73.7 | 3.3$_{\pm0.003}$ | 2.1$_{\pm0.003}$ | 2.0$_{\pm0.000}$ | 2.4$_{\pm0.003}$ | 3.6M | 0.4G |
| EfficientMod-xxs | 76.0 | 3.0$_{\pm0.000}$ | 2.0$_{\pm0.000}$ | 1.9$_{\pm0.003}$ | 2.2$_{\pm0.000}$ | 4.7M | 0.6G |
| MobileNetV2×1.4 | 74.7 | 2.8$_{\pm0.000}$ | 2.0$_{\pm0.003}$ | 1.2$_{\pm0.000}$ | 1.4$_{\pm0.000}$ | 6.1M | 0.6G |
| DeiT-T | 74.5 | 2.7$_{\pm0.003}$ | 2.4$_{\pm0.003}$ | 1.8$_{\pm0.013}$ | 2.2$_{\pm0.003}$ | 5.9M | 1.2G |
| FasterNet-T1 | 76.2 | 3.3$_{\pm0.003}$ | 2.3$_{\pm0.013}$ | 1.9$_{\pm0.003}$ | 2.1$_{\pm0.010}$ | 7.6M | 0.9G |
| EfficientNet-B0 | 77.1 | 3.4$_{\pm0.000}$ | 2.6$_{\pm0.003}$ | 2.0$_{\pm0.010}$ | 2.3$_{\pm0.000}$ | 5.3M | 0.4G |
| MobileOne-S2 | (77.4) | 2.0$_{\pm0.000}$ | 1.7$_{\pm0.003}$ | 1.1$_{\pm0.000}$ | 1.6$_{\pm0.000}$ | 7.8M | 1.3G |
| EdgeViT-XS | 77.5 | 12.1$_{\pm0.070}$ | 5.9$_{\pm0.043}$ | 2.3$_{\pm0.003}$ | 2.6$_{\pm0.000}$ | 6.8M | 1.1G |
| MobileViTv2-1.0 | 78.1 | 5.4$_{\pm0.003}$ | 4.5$_{\pm0.003}$ | 2.9$_{\pm0.003}$ | 3.0$_{\pm0.023}$ | 4.9M | 1.8G |
| EfficientFormerV2-S1 | 77.9 | 4.5$_{\pm0.000}$ | 2.8$_{\pm0.000}$ | 2.5$_{\pm0.003}$ | 3.0$_{\pm0.000}$ | 6.2M | 0.7G |
| EfficientMod-xs | 78.3 | 3.6$_{\pm0.000}$ | 2.5$_{\pm0.003}$ | 2.3$_{\pm0.003}$ | 2.6$_{\pm0.003}$ | 6.6M | 0.8G |
| PoolFormer-s12 | 77.2 | 4.9$_{\pm0.003}$ | 3.9$_{\pm0.010}$ | 2.4$_{\pm0.003}$ | 2.2$_{\pm0.003}$ | 11.9M | 1.8G |
| FasterNet-T2 | 78.9 | 4.3$_{\pm0.043}$ | 3.5$_{\pm0.000}$ | 2.5$_{\pm0.010}$ | 2.9$_{\pm0.013}$ | 15.0M | 1.9G |
| EfficientFormer-L1 | 79.2 | 3.7$_{\pm0.000}$ | 2.8$_{\pm0.010}$ | 1.7$_{\pm0.000}$ | 1.9$_{\pm0.093}$ | 12.3M | 1.3G |
| MobileFormer-508M | 79.3 | 13.6$_{\pm0.030}$ | 11.5$_{\pm0.000}$ | 7.6$_{\pm0.023}$ | 7.8$_{\pm0.005}$ | 14.8M | 0.6G |
| MobileOne-S4 | (79.4) | 4.7$_{\pm0.003}$ | 3.6$_{\pm0.003}$ | 2.3$_{\pm0.003}$ | 2.7$_{\pm0.000}$ | 14.8M | 3.0G |
| MobileViTv2-1.5 | 80.4 | 7.2$_{\pm0.000}$ | 7.0$_{\pm0.053}$ | 3.8$_{\pm0.003}$ | 3.3$_{\pm0.030}$ | 10.6M | 4.1G |
| EdgeViT-S | 81.0 | 20.7$_{\pm0.070}$ | 10.0$_{\pm0.043}$ | 3.8$_{\pm0.003}$ | 4.2$_{\pm0.003}$ | 13.1M | 1.9G |
| EfficientFormerV2-S2 | 80.4 | 7.3$_{\pm0.000}$ | 4.6$_{\pm0.000}$ | 3.8$_{\pm0.163}$ | 4.5$_{\pm0.000}$ | 12.7M | 1.3G |
| EfficientMod-s | 81.0 | 5.5$_{\pm0.000}$ | 3.9$_{\pm0.000}$ | 3.3$_{\pm0.000}$ | 3.8$_{\pm0.010}$ | 12.9M | 1.4 G |

or kernel size. When increasing the width or kernel size, the latency tendency of our EfficientMod is much smoother than MBConv's, suggesting EfficientMod has great potential to be generalized to larger models.

## H  BENCHMARK RESULTS ON MORE GPUS

In addition to the results on the P100 GPU presented in Table 1, we also conducted latency benchmarks on several other GPU instances, including the T4, V100-SXM2, and A100-SXM4-40GB. We observed that there might be some variances even when the GPU types are the same. Therefore, we randomly allocated GPUs from our server and conducted three separate benchmark tests. Table 10 reports the mean and standard deviation values.

As shown in the table, our EfficientMod consistently performs fast on different GPU devices. An interesting finding is that the results of A100 have a higher latency than V100.

Table 11: Latency benchmark for object detection, Instance segmentation, and semantic segmentation.

| Arch. | Backbone | | MS COCO | | | | ADE20K | | |
|---|---|---|---|---|---|---|---|---|---|
| | | Params | $AP^{box}$ | $AP^{mask}$ | Latency$_{(512\times512)}$ | Latency$_{(1333\times800)}$ | Params | mIoU | Latency$_{(512\times512)}$ |
| Conv. | ResNet-18 | 31.2M | 34.0 | 31.2 | 15.8ms | 19.2ms | 15.5M | 32.9 | 8.5ms |
| Pool | PoolF.-S12 | 31.6M | 37.3 | 34.6 | 30.7ms | 66.9ms | 15.7M | 37.2 | 15.7ms |
| Conv. | EfficientMod-s | 32.6M | **42.1** | **38.5** | 28.3ms | 33.6ms | 16.7M | 43.5 | 17.3ms |
| Atten. | PVT-Tiny | 32.9M | 36.7 | 35.1 | 19.6ms | 37.0ms | 17.0M | 35.7 | 11.8ms |
| Hybrid | EfficientF.-L1 | 31.5M | 37.9 | 35.4 | 20.6ms | 29.9ms | 15.6M | 38.9 | 12.3ms |
| Hybrid | PVTv2-B1 | 33.7M | 41.8 | 38.8 | 24.4ms | 41.7ms | 17.8M | 42.5 | - |
| Hybrid | EfficientF.v2-s2 | 32.2M | 43.4 | 39.5 | 47.9ms | 53.0ms | 16.3M | 42.4 | 26.5ms |
| Hybrid | EfficientMod-s | 32.6M | **43.6** | **40.3** | 29.7ms | 48.7ms | 16.7M | **46.0** | 17.9ms |

Firstly, we show that this is a common phenomenon for almost all models, as we can see in the table. Secondly, we observed consistently low GPU utilization for A100, consistently below

| Batch Size | 1 | 2 | 4 | 8 | 16 | 32 | 64 |
|---|---|---|---|---|---|---|---|
| V100-SXM2 | **3.3** | **3.9** | 5.3 | 8.0 | 13.7 | 25.4 | 48.2 |
| A100-SXM4 | 3.7 | **3.9** | **4.2** | **5.7** | **8.9** | **14.6** | **27.2** |

40%, indicating that A100's strong performance is not being fully harnessed. Thirdly, we evaluated batch size 1 to simulate real-world scenarios. When scaling up the batch size, GPU utilization increased, resulting in lower latency for A100, as depicted above (we toke EfficientMod-s as an example). Lastly, we highlight that latency could be influenced by intricate factors that are challenging to debug, including GPU architectures, GPU core numbers, CUDA versions, Operating Systems, *etc*.

# I  LATENCY BENCHMARK ON DOWNSTREAM TASKS

Besides the study on the scalability of EfficientMod in Sec. G, we also explore the latency on real-world downstream tasks. We directly benchmark methods in Table 7 on one A100 GPU (without converting to ONNX format) and report the latency in Table 11.

Clearly, our EfficientMod also exhibits promising efficiency on these tasks. An intriguing observation is that PoolFormer-S12 exhibits the highest latency. This is particularly interesting, considering that the core operation within the network is the pooling operation. We consider two factors could be contributing to this phenomenon: 1) the PoolFormer network architecture might not be optimized for efficiency. 2) pooling operations might not be as highly optimized in CUDA as convolutions (a phenomenon we've also noticed in our backbone design). Additionally, we have observed that as the input resolution increases, the latency gap between Hybrid EfficientMod-s and Conv EfficientMod-s widens. This is attributed to the computational complexity introduced by the Attention Mechanism in our hybrid version. One potential remedy is to reduce computations by downsizing the resolution for the attention block, similar to the approach employed in EfficientFormerV2. However, our Hybrid EfficientMod-s maintains competitive and promising latency results compared to methods like EfficientFormerV2-s2 and other alternatives.

# J  OPTIMIZATION FOR MOBILE DEVICE

As presented in previous results, our EfficientMod mainly focuses on GPU and CPU devices. Next, we explore the optimization for mobile devices. We convert our PyTorch model to a Core ML model using coremltools[1]. We then make use of the iOS application [2] from MobileOne (Chen et al., 2022b) and benchmark latency on an iPhone 13 (iOS version 16.6.1).

We take a pure convolution-based EfficientMod-xxs (without attention module, which achieves 75.3% top-1 accuracy) and compare it with other networks in Table 12. Based on our observation that permute operation is exceptionally time-consuming in CoreML models, we replace the permute + linear layer with a convolutional layer, which is mathematically equal. Our model is able to achieve 75.3% top-1 accuracy at 1.2ms latency on iPhone 13. Inspired by the analysis of normalization layers in EfficientFormer (Li et al., 2022), we further remove all Layer Normalization and add a Batch Normalization layer after each convolutional layer (which can be automatically fused during inference) and re-train the model. By doing so, we reduce the latency to 0.9 ms and achieve a 74.7% top-1 accuracy, which already achieves a promising result. We further slightly adjust the block and

---

[1]https://github.com/apple/coremltools

[2]https://github.com/apple/ml-mobileone/tree/main/ModelBench

| Model | Top-1 | iPhone Latency | Params | FLOPs |
|---|---|---|---|---|
| MobileNetV2×1.0 | 71.8 | 0.9ms | 3.5M | 0.3G |
| FasterNet-T0 | 71.9 | 0.7ms | 3.9M | 0.3G |
| EdgeViT-XXS | 74.4 | 1.8ms | 4.1M | 0.6G |
| MobileOne | 74.6 | 0.8ms | 4.8M | 0.8G |
| MobileViT-XS | 74.8 | 25.8ms | 2.3M | 1.1G |
| EfficientFormerV2-S0 | 73.7 | 0.9ms | 3.6M | 0.4G |
| MobileNetV2×1.4 | 74.7 | 1.1ms | 6.1M | 0.6G |
| DeiT-Tiny | 74.5 | 1.7ms | 5.9M | 1.2G |
| EfficientMod-xxs(conv) | 75.3 | 1.2ms | 4.4M | 0.7G |
| EfficientMod-xxs(conv)♦ | 74.7 | 0.9ms | 4.4M | 0.7G |
| EfficientMod-xxs(conv)▲ | 75.2 | 1.0ms | 4.8M | 0.6G |

Table 12: We benchmark latency on iPhone 13 to explore the optimization on mobile devices. Note: results with strong training tricks (*e.g.*, re-parameterization and distillation) are ignored for fair comparison. "♦" means we remove LN and add a BN after each convolutional layer. "▲" indicates that we slightly adjust the channel and block number for better accuracy.

channel numbers and get a 75.2% accuracy at 1.0ms. The strong performance indicates that our proposed building block also performs gratifyingly on mobile devices.

## K TENTATIVE EXPLANATION TOWARDS THE SUPERIORITY OF MODULATION MECHANISM FOR EFFICIENT NETWORKS

It has been demonstrated that modulation mechanism can enhance performance with almost no additional overhead in works such as Yang et al. (2022); Guo et al. (2023) and in Table 5. However, the reason hidden behind is not fully explored. Here, we **tentatively** explain the superiority of modulation mechanism, and show that modulation mechanism is especially well suited for efficient networks.

Recall the abstracted formula of modulation mechanism in Eq. 4 that $y = p\left(\texttt{ctx}\left(x\right) \odot v\left(x\right)\right)$, it can be simply rewritten as $y = f(x^2)$, where $f(x^2) = \texttt{ctx}\left(x\right) \odot v\left(x\right)$ and we ignore $p\left(\cdot\right)$ since it is a learnable linear projection. Hence, we can recursively give the output of $l$-th layer modulation block with residual by:

$$x_1 = x_0 + f_1(x_0^2), \tag{6}$$

$$x_2 = x_1 + f_2(x_1^2), \tag{7}$$

$$= x_0 + f_1(x_0^2) + f_2(x_0^2) + 2f_2(x_0 * f_1(x_o^2)) + (f_1(x_0^2))^2, \tag{8}$$

$$x_l = a_1 g_1(x_0^1) + a_2 g_2(x_0^2) + a_3 g_3(x_0^3) + \cdots + a_l g_l(x_0^{2^l}), \tag{9}$$

where $l$ indexes the layer, $a_l$ is the weight for each item, $g_l$ indicates the combined function for $l$-th item, and we do not place emphasis on the details of $g_l$. **With only a few blocks, we can easily project the input to a very high dimensional feature space, even infinite-dimensional space**. For instance, with only 10 modulation blocks, we will get a $2^{10}$-dimensional feature space. Hence, we can conclude that $i$) modulation mechanism is able to reduce the requirement of channel number since it can naturally project input feature to very high dimension in a distinct way; $ii$) modulation mechanism does not require a very deep network since several blocks are able to achieve high dimensional space. However, in the case of large models, the substantial width and depth of these models largely offset the benefits of modulation. Hence, we emphasize that the abstracted modulation mechanism is particularly suitable for the design of efficient networks.

Notice that the tentative explanation presented above does not amount to a highly formalized proof. Our future effort will center on a comprehensive and in-depth investigation.

## L  DETAILED ANALYSIS OF EACH DESIGN

**Efficiency of Slimming Modulation Design**   We have integrated an additional MLP layer into the EfficientMod block to validate the efficiency of slimming modulation design. This modification was aimed at assessing the impact of slimming on both performance and computational efficiency. Remarkably, this resulted in a notable reduction in both GPU and CPU latency, with a negligible impact on accuracy. This underlines the effectiveness of our slimming approach in enhancing model efficiency without compromising accuracy.

| Method | Param. | FLOPs | Top-1 | GPU Latency | CPU Latency |
|---|---|---|---|---|---|
| EfficientMod-s-Conv (sperate MLP) | 12.9M | 1.5G | 80.6 | 6.2 ms | 26.2 ms |
| EfficientMod-s-Conv | 12.9M | 1.5G | 80.5 | 5.8 ms | 25.0 ms |

**Efficiency of simplifying Context Modeling**   To further validate the efficiency of our approach in simplifying context modeling, we compared our single kernel size (7x7) implementation against multiple convolutional layers with varying kernel sizes, as the implementation of FocalNet. Experiments are conducted based on EfficientMod-s-Conv variant. Our findings reinforce the superiority of using a single, optimized kernel size. This strategy not only simplifies the model but also achieves a better accuracy-latency trade-off, demonstrating the practicality and effectiveness of our design choice.

| Kernel Sizes | Param. | FLOPs | Top-1 | GPU Latency | CPU Latency |
|---|---|---|---|---|---|
| [3, 3] | 12.7M | 1.4G | 79.7 | 5.8 ms | 28.5 ms |
| [3, 5] | 12.8M | 1.5G | 80.1 | 6.1 ms | 29.0 ms |
| [3, 7] | 12.9M | 1.5G | 80.2 | 6.4 ms | 29.7 ms |
| [5, 5] | 12.9M | 1.5G | 80.2 | 6.3 ms | 29.2 ms |
| [5, 7] | 13.0M | 1.5G | 80.3 | 6.6 ms | 29.8 ms |
| [3, 5 ,7] | 13.1M | 1.5G | 80.5 | 7.2 ms | 32.4 ms |
| [7] | 12.9M | 1.5G | 80.5 | 5.8 ms | 25.0 ms |

**Integrating Attention in EfficientMod**   The introduction of vanilla attention in the last two stages of EfficientMod aimed to improve global representation. We adjusted the block and channel numbers to ensure the parameter count remained comparable between EfficientMod-s-Conv and EfficientMod-s. The results highlight that EfficientMod-s not only shows improved performance but also reduced latency, thereby validating our approach in integrating attention for enhanced efficiency.

| Method | Param. | FLOPs | Top-1 | GPU Latency | CPU Latency |
|---|---|---|---|---|---|
| EfficientMod-s-Conv | 12.9M | 1.5G | 80.5 | 5.8 ms | 25.0 ms |
| EfficientMod-s | 12.9M | 1.4G | 81.0 | 5.5 ms | 23.5 ms |

As shown above, the additional experiments and analyses affirm the distinct contributions and efficacy of each design element in our model, suggesting our EfficientMod can achieve a promising latency-accuracy trade-off.

