# OpenReview forum: "Efficient Modulation for Vision Networks"
_ICLR.cc/2024/Conference — ICLR 2024 poster_

### Official Review · Reviewer_6eCe · 2023-10-28

**Soundness:** 3 good
**Presentation:** 3 good
**Contribution:** 2 fair
**Rating:** 6
**Confidence:** 4

**Summary:**

UPDATE: the rebuttal has answered many of the issues and I have reflected this in the score.


This paper describes an optimized deep learning architecture for vision tasks. It is related to a line of work that utilizes mixtures of transformers and CNNs or adds modulation to CNNs in order to come up with an architecture with high accuracy and low computational complexity and low latency.

**Strengths:**

S: One strength of the paper is the relative simplicity of the architecture compared to some related works.

S: Another strength seems to be good empirical results on selected vision tasks.

S: Even though on the surface the work seems to be incremental improvement over VAN and FocalNets, this paper generalizes them nicely and provides a simpler alternative, which also seems to perform better. Also, any improvement in efficient models for vision tasks is naturally important.

**Weaknesses:**

W: The paper is mainly constructive and experimental in nature. In the appendix there is a tentative explanation that describes that the modulation might lose effectiveness in larger networks. Expanding on this and at the same time showing this in larger networks would make the contribution stronger.

W: One of the most curious things about the paper is the ablation results in Table 5. From there it seems that replacing the modulation with just a regular residual connection (sum) has quite modest performance drop (abs perf drop 1%). Without the multiplication, unless I am mistaken, the architecture reduces to a ResNet with specific hyperparameters and two-path construction. Could the authors discuss this. I think both contributions are valuable, but I am wondering whether it is correct to attribute the performance of the architecture to the modulation since other aspects seem to play even larger role than abs. perf. of 1%.

W: Continuation of the above (without mult). Would the architecture without mult be second best of the architectures compared in Table 2. If so, please add it there.

W: Without mult cont: What would be the performance of the ResNet (no mult) version with the VIT-style attention layers on top?

W: Since this work is most closely related to VAN and FocalNets those works should be at least briefly mentioned in the related work section as well.

W: In Fig 1 and corresponding text, would it make sense to mention where nonlinearities are applied?

**Questions:**

Q: Is Figure 1a) missing the softmax part of the transformer architecture? I think this is one of the key differences to the modulation designs.

Q: In section 3.1, the FC layer seems the same as a 1x1 pointwise convolutional layer. If this is correct, it might be beneficial to mention this.

---

> ### Author Response · Authors · 2023-11-17
> **Rebuttal (part 1)**
>
> Dear Reviewer 6eCe,
>
> We are grateful for your insightful comments and constructive suggestions. We address your concerns and questions as below:
>
> -----
>
>
> **1. Tentative explanation that describes that the modulation might lose effectiveness in larger networks. Expanding on this and at the same time showing this in larger networks would make the contribution stronger.**
>
> **Response**:
>
> **Firstly, we clarify that while modulation is more suitable for efficient networks, but it is still effective in larger networks as well.**
>
>
> As suggested, we developed a larger variant of EfficientMod by increasing its width and adding more blocks. This variant, named EfficientMod-Large, has approximately 25M parameters. We compare it with related baselines:
>
> | Model | Publisher | Parameters | FLOPs | Accuracy |
> |------|---|----|----|----|
> | Swin Trans | ICCV 2021 | 28M | 4.5G | 81.3 |
> | ConvNeXt | CVPR 2022 | 29M | 4.5G | 82.1 |
> | FocalNet-T [3,5] | NeurIPS 2022 | 28.4M | 4.4G | 82.1 |
> | FocalNet-T [3,5,7] | NeurIPS 2022 | 28.6M | 4.5G | 82.3 |
> | VAN-B2 | CVM 2023 | 26.6 | 5.0G | 82.8 |
> | EfficientMod-Large | - | **24.8M** | **3.1G** | **82.9** |
>
> As evident from the table, EfficientMod-Large, while utilizing fewer parameters and significantly fewer FLOPs, achieves superior performance. This outcome, achieved without extensive network design optimization or a fine-tuned training recipe due to limited time, underscores the scalability and efficiency of EfficientMod in larger networks.
>
>
>
> **Secondly, we provide additional details on the tentative explanation that why modulation mechanism can implicitly achieve high dimensions.**
>
>
> Considering the modulation formula as $y=x^2$ (for simplicity, excluding the transformation function in two branches), where $x \in R^d$. This results in:
>
> $y = [x_1^2, x_2^2, x_3^2, ..., x_d^2] \in R^d$. Here, $y = [x_1^2, x_2^2, x_3^2, ..., x_d^2]$ represents a new feature space that is linearly independent of the original input $x= [x_1, x_2, x_3, ..., x_d]$.
>
> With a skip connection, we effectively combine the original $d$-dimensional features with the new $d$-dimensional features, doubling the feature dimension. This is distinct from simply widening the network's width. Through the modulation mechanism, each layer potentially doubles the feature dimension, leading to a significant increase in dimensionality when multiple layers are stacked.
>
> Notice that this is a tentative explanation and hence we placed it in the suppelemntary. A more rigorous analysis and proof would be an exciting avenue for future research, although it may extend beyond the scope of this current work.
>
>
> --------------------
>
> **2. One of the most curious things about the paper is the ablation results in Table 5 ... since other aspects seem to play even larger role than abs. perf. of 1%.**
>
> **Response**:
>
> **We apologize for any confusion caused by Table 5 and appreciate your curiosity regarding its results**. To clarify, the performance variations observed in the table are attributed to different factors. The reduction in performance in top rows is due to a decrease in parameters and FLOPs. However, the change from multiplication to summation in our model, which does not alter the parameters or FLOPs, results in a significant accuracy drop of 1%.
>
> To provide a clearer understanding, we have revised Table 5 to include FLOPs and parameters, and have explicitly indicated which components were removed or altered:
>
>
>
>
> | f(.) | Conv | g(.) | Params. | FLOPs | Acc | Acc drop | Comment |
> |---|---|---|------|----|----|-------|-------|
> | ✓ | | | 11.3M | 1.2G | 72.7 | -7.8 | Remove second FC layer, no sptial interction |
> | | ✓ | | 10.3M | 1.1G | 78.6 | -0.9 | Remove two FC layers |
> | | | ✓ | 11.3M | 1.2G | 72.3 | -8.2 | Remove first FC layer, no sptail interaction |
> | ✓ | ✓ | | 11.6M | 1.3G | 79.8 | -0.7 | Remove second FC layer |
> | | ✓ | ✓ | 11.6M | 1.3G | 79.6 | -0.9 | Remove first FC layer |
> | ✓ | ✓ | ✓ | 12.9M | 1.4G | 80.5 | - | - |
> | mul. | to | sum | 12.9M | 1.4G | 79.5 | -1.0 | Change multiplication to summation |
>
>
> This updated table elucidates the critical role of the multiplication operation in our method. Changing from multiplication to summation, while keeping the parameters and FLOPs constant, leads to a 1.0% decrease in accuracy, a notable gap for efficient models. In fact, this performance gap is larger than that caused by reducing parameters and FLOPs (see row 2, 4 and row 5).  Row 1 and Row 2 means no convlutional layer in the network, hence no spatial interactions, resulting to significant performance drop. The primary objective of this ablation study was to assess the impact of each component in our design individually. Replacing multiplication with summation does transform our network into a ResNet-like architecture, ignoring detailed designs.
>
> -----
>
> We will post rest responses later.

---

> > ### Author Response · Authors · 2023-11-17
> > **Rebuttal (part 2)**
> >
> > **3. Continuation of the above (without mult). Would the architecture without mult be second best of the architectures compared in Table 2. If so, please add it there.**
> >
> > **Response:** Yes, when we replace multiplication with summation, our model achieves an accuracy of 79.5%. We appreciate your suggestion and will include this variant in Table 2 of our paper for a more comprehensive comparison.
> >
> >
> > -------------
> >
> > **4. Since this work is most closely related to VAN and FocalNets those works should be at least briefly mentioned in the related work section as well.**
> >
> >
> > **Response:** Thank you for highlighting the importance of discussing VAN and FocalNet. We have detailed our comparison with these models in Section 3.1 and Table 4, as shown here:
> >
> >
> > | Model            | Top-1 (%)↑ | GPU (ms)↓ | CPU (ms)↓ | Param. | FLOPs |
> > |------------------|------------|-----------|-----------|--------|-------|
> > | VAN-B0           | 75.4       | 4.5       | 16.3      | 4.1M   | 0.9G  |
> > | FocalNet@4M      | 74.5       | 4.2       | 16.8      | 4.6M   | 0.7G  |
> > | EfficientMod-xxs | **76.0**       | **3.0**       | **10.2**      | 4.7M   | 0.6G  |
> >
> >
> > We recognize the value of your suggestion and will further expand our discussion on VAN and FocalNet in the related work section to provide a more thorough context.
> >
> >
> > --------------
> >
> > **5. In Fig 1 and corresponding text, would it make sense to mention where nonlinearities are applied?**
> >
> > **Response:** The activation function is applied after convolutional layers. We will show that in the figure.
> >
> > ------
> >
> > **Question 1: Is Figure 1a) missing the softmax part of the transformer architecture? I think this is one of the key differences to the modulation designs.**
> >
> > **Response:** You are correct in observing the omission of the softmax layer in Figure 1a. This was done for simplicity, and as noted in the caption, activation and normalization layers were also excluded. We will revise the figure to include the softmax layer, ensuring a more accurate representation of the transformer architecture.
> >
> > --------
> >
> > **Question 2: In section 3.1, the FC layer seems the same as a 1x1 pointwise convolutional layer. If this is correct, it might be beneficial to mention this.**
> >
> > **Response:** Yes, FC layer is mathmatically same as 1x1 point-wise convolutional layer. Thanks for your detailed question and we will will make a note of it in the updated manuscript.
> >
> >
> > ---------
> > Thank you for your detailed questions and insightful suggestions. They have been instrumental in enhancing the clarity and depth of our work. If you have any further questions or require additional discussion, we are more than willing to engage. Your feedback is invaluable to us.

---

> > > ### Comment · Reviewer_6eCe · 2023-11-17
> > > **Thank you**
> > >
> > > Thank you for the rebuttal. It addresses many of the concerns I have raised and I have reflected this in the score.

---

> > > > ### Author Response · Authors · 2023-11-17
> > > > **Thank you**
> > > >
> > > > Dear Reviewer 6eCe,
> > > >
> > > > We are  grateful for your acknowledgment of our rebuttal efforts. Your initial comments were instrumental in guiding our refinements and clarifications, and we are heartened to know that our responses have addressed your concerns effectively.
> > > >
> > > > Please do not hesitate to reach out if there are further questions or additional feedback in the future.
> > > >
> > > > Thank you once again for your valuable insights and positive response to our revisions.
> > > >
> > > > Kind regards,
> > > >
> > > > Authors of of Submission 8518

---

### Official Review · Reviewer_Q55t · 2023-10-31

**Soundness:** 2 fair
**Presentation:** 3 good
**Contribution:** 3 good
**Rating:** 8
**Confidence:** 3

**Summary:**

The authors propose a unified convolutional-based building block, EfficientMod, which incorporates favorable properties from both convolution and attention mechanisms. Comparing the prior work shown in Figure 1(b), EfficientMod firstly fuses the FC layers from the MLP and the modulation block to achieve better efficiency, resulting in a unified block. Secondly, EfficientMod includes simplified context modeling, which employs one large-kernel depth-wise convolution layer between two linear projection layers. Extensive experiments and comparisons demonstrate the effectiveness of the proposed method across a range of different tasks (classification, detection, and segmentation).

**Strengths:**

- The proposed method, EfficientMod, is remarkably simple, which could exert a significant influence when deploying a deep model on a resource-limited device.
- The experimental results clearly showcase the effectiveness of EfficientMod in outperforming existing state-of-the-art methods across various tasks (classification, detection, and segmentation).

**Weaknesses:**

- Examining Figure 1, and comparing (b) and (c), the proposed EfficientMod block fuses the MLP on the top and the modulation block as one unified block to improve efficiency. It is conceivable that this might limit performance when using the same number of parameters. The authors should elucidate the principles behind this design, not only from the perspective of efficiency but also in terms of representational ability.
-  Building on the first point, it is imperative to present a comparison between (b) and (c) with the same number of parameters, both in terms of performance and efficiency, and under the same training settings. For example, a comparison between (b), only with fused MLP, and (c).
- Could the authors discuss whether the transformer block can also benefit from the proposed method for efficiency (at least it is feasible to fuse the MLP into one unified block)?
- In Figure 8 and Figure 9, one of the notations in the legend should be 'EfficientMod'.

**Questions:**

My main concern lies with points 1 and 2 in the weaknesses. I look forward to the authors’ response.

---

> ### Author Response · Authors · 2023-11-20
> **Rebuttal**
>
> Dear Reviewer Q55t,
>
> Thank you for your constructive comments, which we have found immensely valuable. Although the MLP fusion is only a small part of EfficientMod, we appreciate your concerns regarding it (weakness 1,2, and 3) and are glad to discuss it in more detail.
>
> ------
>
> **Weakness 1: The authors should elucidate MLP fusion not only from efficiency but also representational ability.**
>
> **Response:** We appreciate your thoughtful recommendations. We examined the effectiveness of slimming the modulation architecture by integrating an extra MLP layer with our EfficientMod block and modifying expansion ratios to retain comparable parameters and FLOPs. That is, the EfficientMod is divided into an EfficientMod block (having an expansion ratio of 1) and an MLP block (with an expansion ratio of (r-1)). Here are the results based on EfficientMod-s-Conv:
>
> | Method | Parameters | FLOPs | Accuracy | GPU Latency | CPU Latency |
> |----|-----|-----|------|-----|----|
> | EfficientMod-s-Conv (sperate MLP) | 12.9M | 1.5G | 80.6 | 6.2 ms | 26.2 ms |
> | EfficientMod-s-Conv | 12.9M | 1.5G | 80.5 (-0.1) | 5.8 ms (-0.4) | 25.0 ms (-1.2) |
>
>
> As shown in the table, the sperating MLP block just have a marginally better representational ability (80.6 vs. 80.5), which is negligible.
> By fusing MLP with the EfficientMod block, we are able to increase the inference speed on the CPU and GPU a lot.
>
>
> -----
>
> **Weakness 2:  It is imperative to present a comparison between (b) and (c) with the same number of parameters, both in terms of performance and efficiency, and under the same training settings. For example, a comparison between (b), only with fused MLP, and (c)**
>
> **Response:**
> Thanks for your suggestions. Fig.1 (b) is a general concept that describes our summarized modulation mechanism, and the inner implementation of context modeling is not limited. We take FocalNet as an example and compare it with our method. Table 4 presented a comparison between FocalNet@4M and our EfficientMod-xxs. Here, we customized a FocalNet variant at 12.7M parameters and compared it with our EfficientMod-s-Conv for a more detailed comparison.  The results are presented here:
>
>
> | Method | Parameters | FLOPs | Accuracy | GPU Latency | CPU Latency |
> |-----|-----|----|---|----|-----|
> | FocalNet@12M (sperate MLP) | 12.7M | 2.1G | 79.4 | 8.6 ms | 45.1 ms |
> | EfficientMod-s-Conv (sperate MLP) | 12.9M | 1.5G | 80.6 | 6.2 ms | 26.2 ms |
> | EfficientMod-s-Conv | 12.9M | 1.5G | 80.5 (-0.1) | 5.8 ms (-0.4) | 25.0 ms (-1.2) |
>
> FocalNet (an instance for Fig.1 (b)) performs much worse than ours EfficientMod-s-Conv, in terms of both latency and performance. When fusing the MLP block to the focal modulation block, we reduce the latency of FocalNet to 7.9 ms and 44.0 ms on GPU and CPU, respectively. However, we didn't train the model successfully, maybe a detailed modification is required.
>
> -----
>
> **Weakness 3: Discuss whether the transformer block can also benefit from the proposed method for efficiency (at least it is feasible to fuse the MLP into one unified block)**
> **Response**: It is feasiable to Transformer block to fuse the MLP into one unified block. We thank the reviewer for the insightful questions, which has a great impact to Transformer-based architectures, not only in efficient networks, but also in large models.
>
> Indeed, there are some endeavors that tried to fuse MLP block to a unified block for faster inference. For example, ViT-22B [1] applies the Attention and MLP blocks in
> parallel for faster inference, instead of sequentially as in the standard Transformer. In detail, the Q, K, V, and first layer in the original MLP block are fused into one layer, the attention-out layer and the second layer in the original MLP block are fused. Also, PaLM [2] also considers this strategy to speed up inference, rewriting the standard formulation of Transformer from $y=x + MLP(LN(x+Attention(LN(x))))$ to $y=x + MLP(LN(x) + Attention(LN(x)))$. As indicated, the unified block results in roughly 15% faster training speed at large scales.
>
> Hence, it is feasible to fuse the MLP block to a unified block for faster inference. Our EfficientMod represents an early exploration of this approach in the realm of efficient networks.
>
>
>
> [1] Dehghani, Mostafa, et al. "Scaling vision transformers to 22 billion parameters." ICML, 2023.
>
> [2] Brown, Tom, et al. "Language models are few-shot learners." NeurIPS, 2020.
>
>
> -------------------
>
> **Weakness 4:  In Fig. 8 and 9, one of the notations should be EfficientMod**
>
> **Response**: Thank you for pointing out this. We will amend the legend to correctly represent 'EfficientMod' (not 'MobileMod') in our revised manuscript.
>
> -----------------
> Again, we extend our deepest gratitude for your insightful questions.
> We hope our responses can address your concerns. Please feel free to share any further questions or concerns if any. We are eager to engage in further discussions and happy to refining our work based on your valuable insights.

---

> > ### Comment · Reviewer_Q55t · 2023-11-22
> >
> > Thank you for the authors' efforts in the rebuttal. Their response has addressed my concerns, and I am going to increase my rating.

---

> > > ### Author Response · Authors · 2023-11-22
> > > **Thank you!**
> > >
> > > Dear Reviewer Q55t,
> > >
> > > Thank you for your insightful comments regarding the fusion of MLP blocks. Your thoughtful analysis has been helpful in enhancing our work, and we have gained valuable insights from your feedback.
> > >
> > > We are grateful for the opportunity to engage in this constructive dialogue and for the significant contributions you have made to our research.
> > >
> > > Thank you once again for your invaluable guidance and support.

---

### Official Review · Reviewer_fvW3 · 2023-11-01

**Soundness:** 3 good
**Presentation:** 3 good
**Contribution:** 3 good
**Rating:** 5
**Confidence:** 5

**Summary:**

This paper proposes an efficient modulation block to build efficient vision networks. The proposed EfficientMod block mainly consists of a simple context modeling design (CTX) with FC and Conv layers. Extensive experiments on image classification, object detection, and semantic segmentation demonstrate that the proposed method achieves strong performance compared with prior methods.

**Strengths:**

1.	The proposed model is simple yet effective.
2.	The proposed model shows strong performance on several benchmarks, including ImageNet, COCO, and ADE20K.

**Weaknesses:**

1.	In Table 2, there are no latency reported for state-of-the-art efficient models.
2.	The proposed method seems simple and more analysis and motivations for the design are needed to understand the principal of the design choice.
3.	Important baselines such as ConvNeXt and Swin Transformer are not included in the comparisons.

**Questions:**

See the weakness part

---

> ### Author Response · Authors · 2023-11-13
> **Rebuttal**
>
> Dear Reviewer fvW3,
>
> Thanks a lot for your insightful feedback and the time you have dedicated to reviewing our work! We address your concerns as follows:
>
> -----
> **Weakness 1: In Table 2, there are no latency reported for state-of-the-art efficient models.**
>
> **Response:**
> Latency (for both CPU and GPU) is primarily reported in Table 1, comparing state-of-the-art efficient models. Table 2 focuses on demonstrating the empirical improvements of introducing attention to our method.
>
> However, to address your concern, we have benchmarked all models in Table 2 (incorporating results from Table 1) and present the updated results here:
>
>
> | Arch.|Model | Params | FLOPs | Acc. | Epoch | Latency GPU | Latency CPU |
> |:--------:|:----:|:------:|:-----:|:----:|:-----:|-----|--------|
> | Conv |RSB-ResNet-18 |12.0M |1.8G | 70.6 |300| 2.0 | 11.5|
> | Conv |RepVGG-A1 |12.8M |2.4G | 74.5 |120| 2.0 | 15.0|
> | Conv | PoolFormer-s12 |11.9M |1.8G | 77.2 |300| 5.0 | 22.3|
> | Conv | GhostNetv2x1.6 |12.3M |0.4G | 77.8 |450| 5.7 | 19.1|
> | Conv |RegNetX-3.2GF |15.3M |3.2G | 78.3 |100| 6.1 | 22.7|
> | Conv |FasterNet-T2|15.0M |1.9G | 78.9 |300| 4.4 | 18.4|
> | Conv |ConvMLP-M |17.4M |3.9G | 79.0 |300| 4.1 | 32.1|
> | Conv | GhostNet-A |11.9M |0.6G | 79.4 |450| - | - |
> | Conv |MobileOne-S4|14.8M |3.0G | 79.4 |300| 4.8 | 26.6|
> | Conv | EfficientMod-s |12.9M |1.5G | **80.5** |300| 5.8 | 25.0|
> | + Atten. | EfficientMod-s |12.9M |1.4G | **81.0** |300| 5.5 | 23.5|
>
>
> As illustrated, our method demonstrates both high speed and promising performance. While a single scale of models may not fully represent the performance-latency trade-off across different models, we invite readers to refer to Figure 3 in our submission for a comprehensive efficiency analysis of our method.
>
>
>
> --------
> **Weakness 2: The proposed method seems simple and more analysis and motivations for the design are needed to understand the principal of the design choice.**
>
> **Response:**
> We appreciate your perspective on the simplicity of our method. We believe that its simplicity, coupled with effectiveness, could be an advantage
> . The primary motivation for our design is the suitability of the modulation mechanism for efficient networks, given its computational efficiency and dynamic nature, blending the benefits of convolution and attention. This motivation of modulation mechanism enables EfficientMod to be straightforward yet impactful.
>
> In detail, we further elaborate on our motivations and the analysis behind our design choices to clarify our approach:
>
> **Principles and Contributions:** At the core of our work, we have summarized the modulation mechanism and further developed the EfficientMod block. This design effectively harnesses the strengths of both convolution (i.e., local representation) and attention mechanisms (dynamics).
>
> **Customization of the EfficientMod Block:** We have tailored the EfficientMod block to enhance efficiency and effectiveness. This customization involved streamlining the modulation design and simplifying the context modeling process. These refinements are also important in achieving best balance between performance and computational efficiency.
>
> **Integration with Attention Blocks:** In integrating attention blocks, we strategically placed attention blocks only in the last two stages of the network for computational efficiency.
>
>
>
>
> --------------------
> **Weakness 3: Important baselines such as ConvNeXt and Swin Transformer are not included in the comparisons.**
>
> **Response:**
> We excluded ConvNeXt and Swin Transformer because they were not designed for efficient networks. However, based on your suggestion, we have carefully designed efficient versions of these models and benchmarked them as follows:
>
>
> | Model| Top-1 | Param. | FLOPs| GPU latency| CPU latency (ms) |
> |---------|-------|--------|--------|--------------|------------------|
> | ConvNeXt @4M |75.5 | 4.3M | 0.7G | 3.1 ms| 11.8 ms |
> | Swin-T @4M |71.1 | 4.8M | 0.8G | -| -|
> | EfficientMod-xxs |**76.0** | 4.7M | 0.6G | **3.0 ms** | **10.2 ms** |
>
> Our EfficientMod-xxs model consistenly outperforms both ConvNeXt and Swin Transformer in terms of latency and accuracy. For ConvNeXt, we reduced the embedding dimension from 96 to 36. For Swin Transformer, we adjusted the configuration to embed_dim=32, depths=[3, 3, 9, 3], and num_heads=[4, 4, 8, 8], resulting in ConvNeXt @4M and Swin-T @4M with computational complexities comparable to our EfficientMod-xxs. These models were trained using the EfficientMod training recipe for a fair comparison. Swin-Transformer cannot be easily exported to ONNX format, but it empirically infers much slow then ConvNeXt and our method.
>
> ------
> In conclusion, we hope that our responses and the additional results address your concerns effectively. We are grateful for the opportunity to improve our work based on your valuable feedback! We are eager to make any additional revisions that might be necessary and are open to further discussion.
>
> Best,
> Authors of Submission 8518

---

> > ### Author Response · Authors · 2023-11-21
> > **Thank you**
> >
> > Dear Reviewer fvW3,
> >
> > We thank you once again for your insightful feedback and the valuable time you have dedicated to reviewing our manuscript.
> >
> > We have addressed each of your concerns in detail, as outlined above, and have conducted additional experiments for weakenss 3. We are open to further discussion and would appreciate any additional comments or suggestions you may have. Your feedback is instrumental in helping us refine our research.
> >
> > Warm regards,
> >
> > Authors of Submission 8518

---

> ### Author Response · Authors · 2023-11-21
> **Further experimental studies for Weakness 2**
>
> We greatly appreciate the insightful feedback from the reviewers.  As Weakness 1 and Weakness 3 are responsed by expiermental results, we have also conducted additional experiments to strengthen our arguments and address the identified weaknesses comprehensively for Weakness 2.
>
> In previous rebuttal, we discussed about the motivation for our design. Here we experimentally show the detailed analysis for our designs.
>
>
>
>
> 1. **Efficiency of Slimming Modulation Design (Sec. 3.2)**.
> In addressing the concerns regarding our slimming modulation design, we have integrated an additional MLP layer into the EfficientMod block. This modification was aimed at assessing the impact of slimming on both performance and computational efficiency. Remarkably, this resulted in a notable reduction in both GPU and CPU latency, with a negligible impact on accuracy. This underlines the effectiveness of our slimming approach in enhancing model efficiency without compromising accuracy.
>
> | Method | Parameters | FLOPs | Accuracy | GPU Latency | CPU Latency |
> |-----------|------|-------|------|-------|-------|
> | EfficientMod-s-Conv (sperate MLP) | 12.9M | 1.5G | 80.6 | 6.2 ms | 26.2 ms |
> | EfficientMod-s-Conv | 12.9M | 1.5G | 80.5 | 5.8 ms (-0.4) | 25.0 ms (-1.2) |
>
>
> 2. **Efficiency of simplifying Context Modeling (Sec. 3.2)**.
> To further validate the efficiency of our approach in simplifying context modeling, we compared our single kernel size (7x7) implementation against multiple convolutional layers with varying kernel sizes, as the implementation of FocalNet. Our findings reinforce the superiority of using a single, optimized kernel size. This strategy not only simplifies the model but also achieves a better accuracy-latency trade-off, demonstrating the practicality and effectiveness of our design choice.
>
> | Method | Parameters | FLOPs | Accuracy | GPU Latency | CPU Latency |
> |----------|-------|-------|------|-------|-------|
> | EfficientMod-s-Conv (kernel: [3, 3]) | 12.7M | 1.4G | 79.7 | 5.8 ms | 28.5 ms |
> | EfficientMod-s-Conv (kernel: [3, 5]) | 12.8M | 1.5G | 80.1 | 6.1 ms | 29.0 ms |
> | EfficientMod-s-Conv (kernel: [3, 7]) | 12.9M | 1.5G | 80.2 | 6.4 ms | 29.7 ms |
> | EfficientMod-s-Conv (kernel: [5,5]) | 12.9M | 1.5G | 80.2 | 6.3 ms | 29.2 ms |
> | EfficientMod-s-Conv (kernel: [5,7]) | 13.0M | 1.5G | 80.3 | 6.6 ms | 29.8 ms |
> | EfficientMod-s-Conv (kernel: [3, 5,7]) | 13.1M | 1.5G | 80.5 | 7.2 ms | 32.4 ms |
> | EfficientMod-s-Conv (kernel: [7], ours) | 12.9M | 1.5G | 80.5 | 5.8 ms | 25.0 ms |
>
>
>
> 3. **Integrating Attention in EfficientMod (Sec. 3.3)**:
> The introduction of vanilla attention in the last two stages of EfficientMod aimed to improve global representation. We adjusted the block and channel numbers to ensure the parameter count remained comparable between EfficientMod-s-Conv and EfficientMod-s. The results highlight that EfficientMod-s not only shows improved performance but also reduced latency, thereby validating our approach in integrating attention for enhanced efficiency.
>
>
> | Method | Parameters | FLOPs | Accuracy | GPU Latency | CPU Latency |
> |----------|--------|-------|----------|-----|--------|
> | EfficientMod-s-Conv | 12.9M | 1.5G | 80.5 | 5.8 ms | 25.0 ms |
> | EfficientMod-s | 12.9M | 1.4G | 81.0 | 5.5 ms | 23.5 ms |
>
> In conclusion, the additional experiments and analyses affirm the distinct contributions and efficacy of each design element in our model. We believe these results comprehensively address the concerns raised and further underscore the novelty and significance of our work.
>
> We thank you for your constructive feedback, which has been instrumental in refining our paper and clarifying our contributions to the field.

---

> > ### Author Response · Authors · 2023-11-22
> > **Thank you and looking forward to further discussions**
> >
> > Dear Reviewer fvW3,
> >
> > We deeply appreciate your thoughtful and constructive review, which has undoubtedly enhanced the quality and clarity of our paper.  We also appreciate your recognition on the soundness, presentation, and contributions in our work.
> >
> > As the rebuttal deadline is approaching, we are eager to know if our responses have sufficiently addressed your concerns or if there are any further clarifications needed from our end. Your feedback is crucial for us to refine our work and contribute to the field.
> >
> > We look forward to hearing your thoughts and are ready to provide any additional information or clarifications that may be required.
> >
> > Thanks,
> >
> > Authors of Submission 8518

---

> > > ### Comment · Reviewer_fvW3 · 2023-11-22
> > > **Thanks for the response**
> > >
> > > Thanks for the detailed response. Overall some of my previous concerns have been addressed, but I tend to keep my rating. In the above rebuttal, the authors compare the proposed method with ConvNeXt @4M	model with 75.5\% top1 accuracy and 0.7G FLOPs. However, the official paper actually releases ConvNeXt/ConvNeXtv2 models with small scales, as listed in https://github.com/facebookresearch/ConvNeXt-V2#imagenet-1k-fine-tuned-models. Also, as shown in Fig. 1 of ConvNeXt-V2 paper, ConvNeXt-Atto and ConvNeXt-Femto has 75.7\% and 77.5\% top1 accuracy with 0.55G and 0.78G FLOPs, which might be better than the proposed method.

---

> > > > ### Author Response · Authors · 2023-11-22
> > > > **Thank you! Comparing to results in ConvNextV2, EfficientMod is better**
> > > >
> > > > Dear Reviewer fvW3,
> > > >
> > > > Thank you for your valuable feedback and for acknowledging that our rebuttal has addressed some of your concerns. We would like to further clarify the performance comparison between our EfficientMod and the ConvNeXt variants (Atto, Femto) as detailed in the ConvNeXtv2 paper (**EfficientMod performs better**):
> > > >
> > > > **With ~0.6G FLOPs**: ConvNeXt-Atto achieves a top-1 accuracy of 75.7%, whereas our EfficientMod shows a slightly higher accuracy of 76.0%, **we perform 0.3% better**.
> > > >
> > > > **With ~0.8G FLOPs**: ConvNeXt-Femto reaches 77.5% top-1 accuracy, whereas our EfficientMod achieves 78.3%, **we perform 0.8% better**.
> > > >
> > > > Additionally, to ensure a fair comparison, **the ConvNeXt@4M model was trained using the same training recipe as our EfficientMod**. We costumized ConvNeXt@4M by decreasing the dimension number of  ConvNeXt-tiny from [96, 192, 384, 768] to [36, 72, 144, 288], and keeping all other configurations same.  Due to the limited time and our nodes allocation limitation, we can not train these models (atto and femto) using our training recipe and further benchmark the latency.
> > > >
> > > > **We hope this detailed comparison helps in resolving any misunderstandings regarding the performance gap between the ConvNeXt variants and our work**.
> > > >
> > > >
> > > > Thank you once again for your time and constructive feedback.

---

> > > > ### Author Response · Authors · 2023-11-23
> > > > **Thank you! We are better than results in ConvNeXtv2 paper**
> > > >
> > > > Dear Reviewer fvW3,
> > > >
> > > > Thank you for your insightful comments on our work. We'd like to clarify that **our results actually outperform those in the ConvNextv2 paper** by 0.3% at 0.6G FLOPs and 0.8% at 0.8G FLOPs. Given this and your acknowledgment that we've addressed your concerns, could you kindly reconsider the score for our submission?
> > > >
> > > > Best,
> > > >
> > > > Authors of Submission 8518

---

> ### Author Response · Authors · 2023-11-22
> **Looking forward to further discussions**
>
> Dear Reviewer  fvW3,
>
> Thank you for the time and effort you have devoted to reviewing our work. We have carefully considered and responded in detail to the questions and concerns you raised.
>
> As the author-reviewer discussion deadline is drawing near, we would like to know if our responses have effectively addressed your concerns. Your insights are invaluable to us, and we aim to ensure that our paper meets the highest standards based on your feedback.
>
> We would like to again appreciate for your time for helping strengthen the paper with your suggestions.
>
> Looking forward to your feedback.

---

### Official Review · Reviewer_xsDm · 2023-11-02

**Soundness:** 2 fair
**Presentation:** 2 fair
**Contribution:** 2 fair
**Rating:** 5
**Confidence:** 4

**Summary:**

This paper introduced a new model structure based on previous modulation designs to further improve the efficiency (especially inference latency) and performance. The paper revisited previous modulation designs and improved the efficiency by reducing fragmented operations and simplifying the structure. The proposed method shows better performance than previous efficient networks on ImageNet with lower latency. The improvements also transfer to detection and segmentation.

**Strengths:**

1.	The paper had a clear introduction to previous works and how is the proposed method motivated from these works. This makes it easier to follow the work and understand how it works.
2.	There are extensive experiments on multiple tasks. And the proposed method achieves better performance and latency than previous efficient models.

**Weaknesses:**

1.	There are limited technical contributions in the work. This paper focuses on improving the latency of previous works. The improvements/changes from previous works are mainly engineering designs, for example, fuse multiple FC layers together, fuse multiple DWConv into a larger one, replace reshape operation with repeat. The guidance is mainly from previous works such as ShuffleNet v2, which is to reduce fragmented operations for improved latency. There are limited new insights.
2.	It is not clear how much efficiency improvement does each design contribute. I suggest the author to conduct a thorough ablation study to show the impact of each structure change, and explain why it could achieve improvement.
3.	Fig 1 is good to illustrate the difference between the proposed method and previous works. But it could be better to expand Fig 1 (c) in details when explaining the method. This makes it easier to understand the proposed structure and details.
4.	In Table 1, why VAN and FocalNet results are not included? They seem to be the most relevant works.
5.	In Table 2, why adding Attention even reduced the FLOPs?
6.	In Table 1, are the GPU and CPU latency of different models measured on the same device?

**Questions:**

Please see the weakness part

---

> ### Author Response · Authors · 2023-11-11
> **Rebuttal  (part 1)**
>
> Dear Reviewer xsDm,
>
> We greatly appreciate your insightful feedback and the time you have dedicated to reviewing our work! Your comments have been instrumental in enhancing the quality of our research. We address your concerns as follows:
>
> --------
> **Q6: In Table 1, are the GPU and CPU latency of different models measured on the same device?**
>
> **Response:** Yes, all models were benchmarked on the same device to ensure consistency in our measurements.
>
> Additionally, we conducted benchmarks across various GPU types, as detailed in Supplementary Table 10. This approach was taken to maintain uniformity in the testing environment and minimize latency variances across different GPU models.
>
> --------
> **Q5: In Table 2, why adding Attention even reduced the FLOPs?**
>
> **Response:** The reason is that we replaced some EfficientMod blocks with Attention blocks to keep almost the same number of parameters, for a fair comparison.
>
> This involved adjusting the expansion ratio in Attention blocks and other hyper-parameters. We believe that the number of parameters plays a crucial role in performance, and maintaining a consistent parameter number is essential for a fair comparison.
>
>
> -----
> **Q4: In Table 1, why VAN and FocalNet results are not included? They seem to be the most relevant works.**
>
> **Response:**  The comparison (EfficientMod, VAN, and FocalNet) is detailed in our Ablation Study Table 4, rather than Table 1. This is because VAN and FocalNet models do not fall under the category of 'efficient models' and lack small variants.
>
> For the study, we developed a FocalNet variant with 4M parameters (FocalNet@4M), selecting the best-performing variant among several tested. Our analysis in Table 4 demonstrates that EfficientMod not only outperforms FocalNet and VAN in terms of efficiency but also exhibits significantly lower latency on both CPU and GPU.
>
> For your convenience，we included Table 4 in our submission here:
>
> | Model | Top-1(%)↑| GPU (ms)↓ | CPU (ms)↓   | Param. | FLOPs |
> |:------:|---------|------|-------|--------|-------|
> |VAN-B0 | 75.4  | 4.5  | 16.3 | 4.1M   | 0.9G  |
> |FocalNet@4M  | 74.5 | 4.2 | 16.8 | 4.6M   | 0.7G  |
> | EfficientMod-xxs  | **76.0**     | **3.0** | **10.2**  | 4.7M   | 0.6G  |
>
> ---------
> We will be posting our responses to the remaining questions shortly. We look forward to receiving further feedback and engaging in more detailed discussions. Thank you once again for your valuable contributions to improving our research.

---

> ### Comment · Reviewer_xsDm · 2023-11-19
> **Didn't see Part 2**
>
> Thanks for the rebuttal. But I didn't see the part 2 (if there is).

---

> > ### Author Response · Authors · 2023-11-19
> > **Finalizing Part 2**
> >
> > Dear Reviewer xsDm,
> >
> > Thank you for your attention. We are currently finalizing Part 2, which involves incorporating results from extensive ongoing experiments, particularly training several models on ImageNet for Question 2.
> >
> > Meanwhile, if you have any questions or concerns about Part I, please feel free to let us know.
> >
> > We appreciate your patience and look forward to sharing the detailed Part 2 shortly. Thank you again for your insightful questions and constructive suggestions!
> >
> > Kind regards,
> >
> > Authors of Submission 8518

---

> ### Author Response · Authors · 2023-11-20
> **Rebuttal (part 2)**
>
> Dear Reviewer  xsDm,
>
> Thanks for your patience since we are working for the experimental results for rebuttal Q2. Here is our rebuttal part 2.
>
> -------
>
> **Q3: It could be better to expand Fig 1 (c) in details when explaining the method.**
>
> **Response**: Thank you for your kind suggestions,  we would revise the Fig 1 (c) and add more details in it, ensuring that our method is communicated more effectively and comprehensibly.
>
> ----------
>
> **Q2: It is not clear how much efficiency improvement does each design contribute.**
>
> **Response**:
>
> Thank you for prompting further exploration into the efficiency improvements contributed by each design aspect. We have conducted additional experiments to elucidate this:
>
>
> 1. **About the efficiency of Sliming Modulation Design as described in Sec. 3.2**. We investigated the efficiency of slimming the modulation design by integrating an additional MLP layer with our EfficientMod block, adjusting expansion ratios to maintain similar parameters and FLOPs. The results indicate a notable reduction in latency with minimal impact on accuracy, demonstrating the effectiveness of this slimming approach:
>
>
> | Method | Parameters | FLOPs | Accuracy | GPU Latency | CPU Latency |
> |-----------|------|-------|------|-------|-------|
> | EfficientMod-s-Conv (sperate MLP) | 12.9M | 1.5G | 80.6 | 6.2 ms | 26.2 ms |
> | EfficientMod-s-Conv | 12.9M | 1.5G | 80.5 | 5.8 ms (-0.4) | 25.0 ms (-1.2) |
>
>
> 2. **About the efficiency of Simplifying Context Modeling as described in Sec. 3.2**. We compared our implementation using a single kernel size (7x7) with multiple convolutional layers of varying kernel sizes, following FocalNet's context modeling approach. Our results show that adopting a single constructive kernel size optimizes the accuracy-latency trade-off, as detailed in the following table:
>
> | Method | Parameters | FLOPs | Accuracy | GPU Latency | CPU Latency |
> |----------|-------|-------|------|-------|-------|
> | EfficientMod-s-Conv (kernel: [3, 3]) | 12.7M | 1.4G | 79.7 | 5.8 ms | 28.5 ms |
> | EfficientMod-s-Conv (kernel: [3, 5]) | 12.8M | 1.5G | 80.1 | 6.1 ms | 29.0 ms |
> | EfficientMod-s-Conv (kernel: [3, 7]) | 12.9M | 1.5G | 80.2 | 6.4 ms | 29.7 ms |
> | EfficientMod-s-Conv (kernel: [5,5]) | 12.9M | 1.5G | 80.2 | 6.3 ms | 29.2 ms |
> | EfficientMod-s-Conv (kernel: [5,7]) | 13.0M | 1.5G | 80.3 | 6.6 ms | 29.8 ms |
> | EfficientMod-s-Conv (kernel: [3, 5,7]) | 13.1M | 1.5G | 80.5 | 7.2 ms | 32.4 ms |
> | EfficientMod-s-Conv (kernel: [7], ours) | 12.9M | 1.5G | 80.5 | 5.8 ms | 25.0 ms |
>
>
>
> 3. **Integrating Attention in EfficientMod as described in Sec. 3.3**: To enhance global representation, we introduced vanilla attention in the last two stages of EfficientMod. The adjustments of block and channel number ensured comparable parameters between EfficientMod-s-Conv and EfficientMod-s, with the latter showing improved performance and reduced latency:
>
>
> | Method | Parameters | FLOPs | Accuracy | GPU Latency | CPU Latency |
> |----------|--------|-------|----------|-----|--------|
> | EfficientMod-s-Conv | 12.9M | 1.5G | 80.5 | 5.8 ms | 25.0 ms |
> | EfficientMod-s | 12.9M | 1.4G | 81.0 | 5.5 ms | 23.5 ms |
>
> These findings affirm the distinct contributions of each design element to the model's overall efficiency.
>
> --------
>
> **Q1: Limited technical contributions in the work.**
>
> **Response**: We understand your concerns regarding the perceived technical contributions. While certain implementation aspects like layer fusion and reshaping operations are not our main focus, we emphasize that our primary contributions lie in:
>
> **Summarization of Modulation Mechanism**: We provide a comprehensive overview of the modulation mechanism in Section 3.1, highlighting its unique position in the landscape of efficient network designs.
>
> **Efficient Modulation Design**: As detailed in Section 3.2, we introduce the EfficientMod block, a novel building block for efficient networks, leveraging both convolutional and attention mechanisms.
>
>
> Compared with other efficient network designs, we hold the idea that modulation is unique and distinguished. It can take advantage of both convolution (local representation, efficiency) and attention mechanism (dynamics). Here we list a taxonomy of the representative efficient networks in Table 1 based on their main technical contributions:
>
> | Technical Contributions | Networks |
> |-------|----------|
> | DW-Convolution | MobileNet, MobileNetv2 |
> | Channel Shuffle | ShuffleNet, ShuffleNetv2 |
> | Feature Re-use | FasterNet |
> | Re-parameterization | MobileOne |
> | Neural Architecure Searching | EfficientNet, EfficientFormerV2 |
> | Hybrid architecture | MobileViT, EfficientFormerV2, MobileFormer |
> | Modulation Mechanism | EfficientMod (ours) |
>
> This table underscores the distinctiveness of our contribution, positioning the modulation mechanism as a key innovation in EfficientMod. Our work focuses on advancing this aspect, offering a significant step forward in the field of efficient network design.

---

> ### Author Response · Authors · 2023-11-22
> **Thank you!**
>
> Dear Reviewer xsDm,
>
> Thank you very much for your insightful review. Your feedback has been invaluable in helping us enhance the quality of our work.
>
> In response to your concerns, we have provided detailed additional results, further expanded our analysis, and revised our manuscript.
>
> As the rebuttal deadline is approaching, we would greatly appreciate any further comments or suggestions you might have. Could you please let us know if our responses and revisions effectively address your concerns? We are committed to continuous improvement and eagerly await your guidance to further refine our work.
>
> Thank you once again for your constructive feedback and support.
>
> Best regards,
>
> Authors of Submission 8518

---

> ### Author Response · Authors · 2023-11-22
> **Thank you!**
>
> Dear Reviewer xsDm,
>
> We greatly appreciate your constructive suggestions and insightful comments on our work. As the rebuttal deadline is drawing near, we are keen to know if our responses have adequately addressed your concerns.
>
> Please feel free to share any further questions or insights you may have. Your feedback is invaluable to us.

---

> ### Author Response · Authors · 2023-11-22
> **Thank you and looking forward to further feedback**
>
> Dear Reviewer xsDm,
>
> We sincerely appreciate your insightful comments on our submission (ID: 8518). As the review period is nearing its conclusion, we are eager to know if our rebuttal has effectively addressed your concerns and questions.
>
> Should you have any further questions, or if there are additional aspects of our work that you would like us to clarify, please do not hesitate to let us know. Your feedback is invaluable to us!
>
> Best regards,
>
> Authors of Submission 8518

---

> > ### Comment · Reviewer_xsDm · 2023-11-23
> > **Thanks for the rebuttal**
> >
> > Thanks for the rebuttal. After reading the rebuttal, I still think there are very limited insightful takeaways and analyses in terms of the design pricinples (Reviewer fvW3 also shared this concern). Thus, I would like to keep my score.

---

> ### Author Response · Authors · 2023-11-23
> **Thanks for the feedback and our clarification**
>
> Dear Reviewer xsDm,
>
> Thanks for your feedback. We believe that in our paper and rebuttal, we have provided detailed analyses and enough insightful takeaways. To ease your understanding, we summarize as follows:
>
> --------
> **insightful takeaways:**
>
> 1. **We summarized the modulation mechanism**, and we pointed out that the modulation mechanism could be the most important part other than detailed implementations in related work like VAN and FocalNet. The modulation mechanism brings advantages from bot convolution (local receptive field, efficiency, etc) and attention (dynamics).
>
> 2. **We introduced the EfficientMod block based on the modulation mechanism**,  experimental results indicated that our EfficientMod block can serve as an essential building block for efficient networks, supported by Table 1 and Table 6.
>
> In details of  the  EfficientMod design, we discuss two points:
>
> 1. "Sliming Modulation Design by fusing MLP block, generating a unified block". By doing so, we can achieve comparable performance and speed up inference a lot, as supported by our rebuttal. Also, Reviewer Q55t is really interesting in this part and increases the score to 8.
>
> 2. "Simplifying Context Modeling by keeping the most constructive component in the design." By doing so, we show that we don't need sophisticated designs (which are also not good for efficiency), a block that inherits advantages from the modulation mechanism can achieve promising performance.
>
> -----------------------------------
> **analyses in terms of the design principles:**
>
> 1. **We have detailed the analyses of design principles in our rebuttal 2.** This is also acknowledged by Reviewer Q55t, who raised the score to 8.
>
> 2. **We have presented detailed ablation studies in Section 4.2 ablation studies, and in Table 5.** Table 5 is also questioned by Reviewer 6eCe; after our rebuttal, Reviewer 6eCe increased his score to 6.
>
> -----------------------------
> Most importantly, we believe that our main contribution "summarization of modulation mechanism" and "EfficientMod block" should not be overlooked.  **With modulation mechanism, we can improve the performance by 1.0% on ImageNet, WITHOUT ANY additional overhead,** as shown in Table 5 last row.
>
> Thanks a lot for your time. We hope our clarification can help you better understand our work.

---

### Author Response · Authors · 2023-11-21
**Thank you! Summary of our manuscript modifications**

**Dear all reviewers**,

Thanks a lot for your constructive suggestions and insightful comments. We have made the following revisions to our manuscript:

--------

1. As recommended by Reviewer 6eCe, we updated the figures in Fig. 1 and added softmax in Transformer block (a). As suggested by Reviewer xsDm, we added more details, such as activation and dimension number, in our EfficientMod block (c).


2. Thanks to Reviewer Q55t, we revised the notations in the legend in Fig. 8 and Fig. 9.


3. In addition, as Reviewer 6eCe suggested, we mentioned FocalNet and VAN in related work.


4. As indicated by Reviewer fvW3 and Reviewer Q55t, we we have included a thorough analysis of each design's performance and efficiency in the supplementary Sec. L.

---------

We are grateful for the opportunity to improve our work based on your valuable feedback! We are eager to make any additional revisions that might be necessary and are open to further discussion.



Kind regards,

Authors of Submission 8518

---

### Author Response · Authors · 2023-11-23
**Thank you and some clarifications.**

Dear all reviewers,

We deeply appreciate your valuable suggestions and  insightful comments.

------

A special thank you to Reviewer 6eCe for recognizing our efforts in the rebuttal and increasing the score. Your detailed questions have greatly improved our work.

Thanks to Reviewer Q55t for acknowledging our analysis in MLP block fusion and for the positive adjustment in scoring. Your invaluable insights helped us a lot and we have learnt a lot from the discussions.

We also appreciate Reviewer fvW3's additional feedback. We hope our further clarification on EfficientMod's performance over ConvNeXt has been helpful and can avoid  misunderstandings.

Thank you, Reviewer xsDm, for the engaging discussion on our technical contributions. We hope our clarifications on the main contributions (summarization of modulation mechanism and the derived EfficientMod block) have enhanced the understanding of our work.

-----
Happy Thanksgiving Day!

Best,

Authors of Submission 8518

---

### Meta-Review · Area_Chair_tEbh · 2023-12-05

**Metareview:**

The paper introduces an efficient network design based on a modulation mechanism that inherits the merits of convolution (local receptive fields, efficiency) and self-attention (global context). The key insights and design motivation is that competing methods have fragmented FC operation or successive application of depth-wise convolution, which reduces efficiency/speed. This work aims at fusing the former and replacing the latter with a single large kernel convolution. Overall, the paper is well organized, clearly written. The proposed technique is simple, motivated, can be used as a drop-in replacement, and leads to strong performance on various benchmarks. The majority of the benchmark is done for classification, but the design also leads to strong performance on segmentation and detection benchmarks. Another sign of its potential for efficient design in resource-constrained devices/systems. All of the above strengths are unanimously praised by reviewers.

**Justification For Why Not Higher Score:**

The very high-level ideas already exist in the literature for instance ShuffleNet. This work combines many existing insights, and proposed an effective mechanism that's well motivated, and fairly validated.

**Justification For Why Not Lower Score:**

The paper received 1x accept, 1x slightly above acceptance, and 2x slightly below acceptance. Having read the paper, and the weaknesses and rebuttals, I recommend accepting this paper as a poster. Most of the weaknesses for the 2x below acceptance have been addressed in the rebuttal. I believe the remaining concerns on experimental design choice on ConvNeXt are acceptable, the authors make the effort to fairly compare the methods under the same parameter count, and the same training settings.

---

### Decision · Program_Chairs · 2024-01-16

Accept (poster)